# Hexameric NuMA:LGN structures promote multivalent interactions required for planar epithelial divisions

Laura Pirovano[1,6], Simone Culurgioni[1,3,6], Manuel Carminati [1,4,6], Andrea Alfieri [1,5,6], Silvia Monzani[1], Valentina Cecatiello[1], Chiara Gaddoni[1], Francesca Rizzelli[1], James Foadi [2], Sebastiano Pasqualato [1] & Marina Mapelli[1]

Cortical force generators connect epithelial polarity sites with astral microtubules, allowing dynein movement to orient the mitotic spindle as astral microtubules depolymerize. Complexes of the LGN and NuMA proteins, fundamental components of force generators, are recruited to the cortex by Gαi-subunits of heterotrimeric G-proteins. They associate with dynein/dynactin and activate the motor activity pulling on astral microtubules. The architecture of cortical force generators is unknown. Here we report the crystal structure of NuMA:LGN hetero-hexamers, and unveil their role in promoting the assembly of active cortical dynein/dynactin motors that are required in orchestrating oriented divisions in polarized cells. Our work elucidates the basis for the structural organization of essential spindle orientation motors.

[1] IEO, European Institute of Oncology IRCCS, 20141 MILANO, Italy. [2] Department of Mathematical Sciences, University of Bath, Claverton Down, Bath BA2 7AY, UK. [3] Present address: Exscientia Ltd., The Schröedinger Building, Heatley Road, Oxford Science Park, Oxford OX4 4GE, UK. [4] Present address: MRC Laboratory of Molecular Biology, Cambridge CB2 0QH, UK. [5] Present address: Department of Biosciences, Università degli Studi di Milano, 20133 Milan, Italy. [6] These authors contributed equally: Laura Pirovano, Simone Culurgioni, Manuel Carminati, Andrea Alfieri. Correspondence and requests for materials should be addressed to M.M. (email: marina.mapelli@ieo.it)

In multicellular organisms, oriented cell divisions sustain tissue morphogenesis and homeostasis by ensuring the correct positioning of daughter cells after cytokinesis[1]. Accurate execution of oriented divisions relies on the positioning of the mitotic spindle, which is attained in metaphase by the action of molecular motors coordinated with cortical polarity cues.

The major molecular motor responsible for spindle positioning is cytoplasmic dynein 1 (hereon referred to as dynein), a multi-subunit AAA-type ATPase that works with dynactin and moves towards the minus-end of microtubules[2]. The motility of dynein/dynactin is activated by cargo adaptors, including BicD2[3–5], and Hook3[6]. The most credited model assumes that cortical NuMA targets and anchors dynein/dynactin to the plasma membrane so that its retrograde motor activity on dynamic astral microtubules (MTs) results in traction forces pulling the spindle toward the cortex[7,8]. How the coordinated action of these cortically-localized force generating machines orchestrates spindle placement remains poorly understood.

NuMA is a 250 kDa nuclear protein released in the cytoplasm after nuclear envelope break-down. The first 705 residues of NuMA are sufficient to immuno-precipitate dynein/dynactin from mitotic lysates[7]. Interestingly, light-induced ectopic delivery of this NuMA fragment to the cortex results in dynein/dynactin recruitment but cannot support spindle pulling[8], implying that during spindle orientation additional functions encoded by the coiled-coil region and the C-terminal cargo binding portion of NuMA are essential for the spindle orientation process.

Overall, the domain structure of NuMA consists of a globular N-terminal domain predicted to fold as a Hook domain, a central 1500-residue long coiled-coil mediating self-assembly, and an unstructured C-terminal portion harboring bindings sites for microtubules[9–11], lipids[12,13], importin-α[14], the cortical protein 4.1R[15,16], and the spindle orientation protein LGN[17,18] (Supplementary Fig. 1). How these NuMA binding partners contribute to the cortical localization of the protein has been extensively studied. In metaphase, when spindle positioning takes place, NuMA accumulates at the cortex by association with LGN. LGN in turn is targeted to the plasma membrane by direct interaction with multiple copies of the Gαi subunit of heterotrimeric G-proteins inserting a myristoyl group in the lipid bilayer[18]. Depolymerization of astral MTs by low doses of nocodazole does not affect the cortical localization of NuMA in metaphase (unpublished data) in spite of causing spindle misorientation[8]. These observations suggest that NuMA is not transported to the cortex along astral MTs by kinesins, but rather acts upstream of the microtubule motors in the assembly of cortical force generating complexes. The functional role of the MT-binding activity of NuMA is poorly understood. Two discontinuous regions of NuMA associate with MTs: the first one spans residues 1914–1985 and is incompatible with LGN-binding[10], while the second lies in the C-terminal tail after residue 2001, is compatible with LGN binding[9], and has recently been implicated in spindle placement[8]. Intriguingly, we previously found that phosphorylation of Ser2047 of NuMA by Aurora-A regulates cortical NuMA recruitment, for reasons that are still unclear[9]. Two positively-charged stretches of NuMA upstream and downstream of the LGN-binding domain associate with PIP2 at the plasma membrane. Being negatively regulated by Cdk1 phosphorylation, this interaction is not involved in spindle placement as in early mitosis[12,13]. However, PIP2-binding promotes an additional cortical NuMA accumulation occurring in anaphase, upon Cdk1 inactivation, that supports spindle elongation and sister chromatid separation[12,15,16].

The overall three-dimensional organization of active force generators is poorly understood. Ectopic targeting experiments indicate that clustering of NuMA at the cell cortex may be required for efficient pulling on astral microtubules[8], although the origin of the multivalent interactions required for cluster formation is unknown. Structural studies revealed that the elongated NuMA peptide encompassing residues 1900–1928 lines the inner side of the helical N-terminal TPR scaffold of LGN, engaging in a nanomolar affinity interaction[19,20]. The TPR domain of LGN interacting with NuMA comprises 8 tetratricopeptide repeats (TPRs), each consisting of a couple of antiparallel helices organized in a concave super-helical array, in such a way that the first helix-A of each TPR faces the inner side of the super-helix, while the second helix-B is positioned outside. A peculiar feature of the TPR domain of LGN is the presence of flanking extensions at the N-terminus and C-terminus which are predicted to adopt a helical conformation. The flexible C-terminal portion of LGN associates cooperatively with four Gαi molecules. These NuMA/LGN/Gαi assemblies constitute core modules connecting dynein/dynactin and MTs plus-TIPs to the cortex.

Here, we report the crystallographic structure of NuMA:LGN hetero-hexamers assembled on a NuMA fragment longer than the minimal binding peptide. We show that these high-order oligomers are required for spindle orientation in an epithelial model. Molecularly, these hexameric complexes in combination with dimeric full-length NuMA can generate extended cortical protein networks that spatially organize dynein/dynactin on astral MTs to position the spindle.

## Results

**Structural analysis of NuMA/LGN hexamers**. To start investigating the architecture of force generators, we reconstituted the NuMA:LGN interaction using human proteins expressed in bacteria, starting from the previously identified binding interfaces[20,21]. Size-exclusion chromatography (SEC) elution profiles of LGN[1–409] (LGN$^{TPR}$ in the following) bound to NuMA[1900–1928] were consistent with a 1:1 binary interaction, in agreement with the available structural data[19] (Fig. 1a, b). However, the longer NuMA fragment encompassing residues 1821–2001 assembles with LGN$^{TPR}$ to form a complex that elutes much earlier than its expected molecular weight (Fig. 1b). We reasoned that this longer NuMA fragment could form high-order oligomers with the TPR domain of LGN. Static-Light-Scattering (SLS) analysis confirmed that NuMA[1821–1928] and LGN$^{TPR}$ form hetero-hexamers (Fig. 1c). We then set out to map the molecular requirements for the NuMA:LGN oligomerization by performing SEC and SLS assays on a battery of truncated LGN and NuMA mutants (Fig. 1c, d). Our analysis revealed that NuMA[1861–1928] and LGN[7–367] are the minimal constructs interacting with a 3:3 stoichiometry (Fig. 1e), and any shortening at the N-terminal or C-terminal ends results in a 1:1 complex (Fig. 1c, d). Thus, in the following, we will refer to NuMA[1861–1928] as NuMA$^{LGNBD}$.

**Architecture of NuMA/LGN hexamers**. To gain insights into the topology of the NuMA$^{LGNBD}$:LGN$^{TPR}$ hetero-hexamers, we determined their crystallographic structure at 4.3 Å resolution. As diffraction from individual crystals was very weak, a multi-crystal approach was adopted. Several datasets were collected from multiple crystals and merged to produce a combined dataset with 99.9% completeness and a I/sigma of 19.1 (CC$_{1/2}$ of 0.99) (see the Methods section for details). The structure was then solved by molecular replacement using NuMA[1900–1928]:LGN$^{TPR}$ as a search model, and refined to an R$_{free}$ of 23.3% and R$_{work}$ of 18.2%, with good stereochemistry (Table 1). The final model includes residues 7–367 of LGN, and residues 1864–1928 of NuMA with gap-spanning residues 1881–1897.

Overall, the hetero-hexamers arrange in a donut-shaped architecture, with the backbone of the donut formed by the three LGN$^{TPR}$ protomers concatenated in a head-to-tail fashion, and a

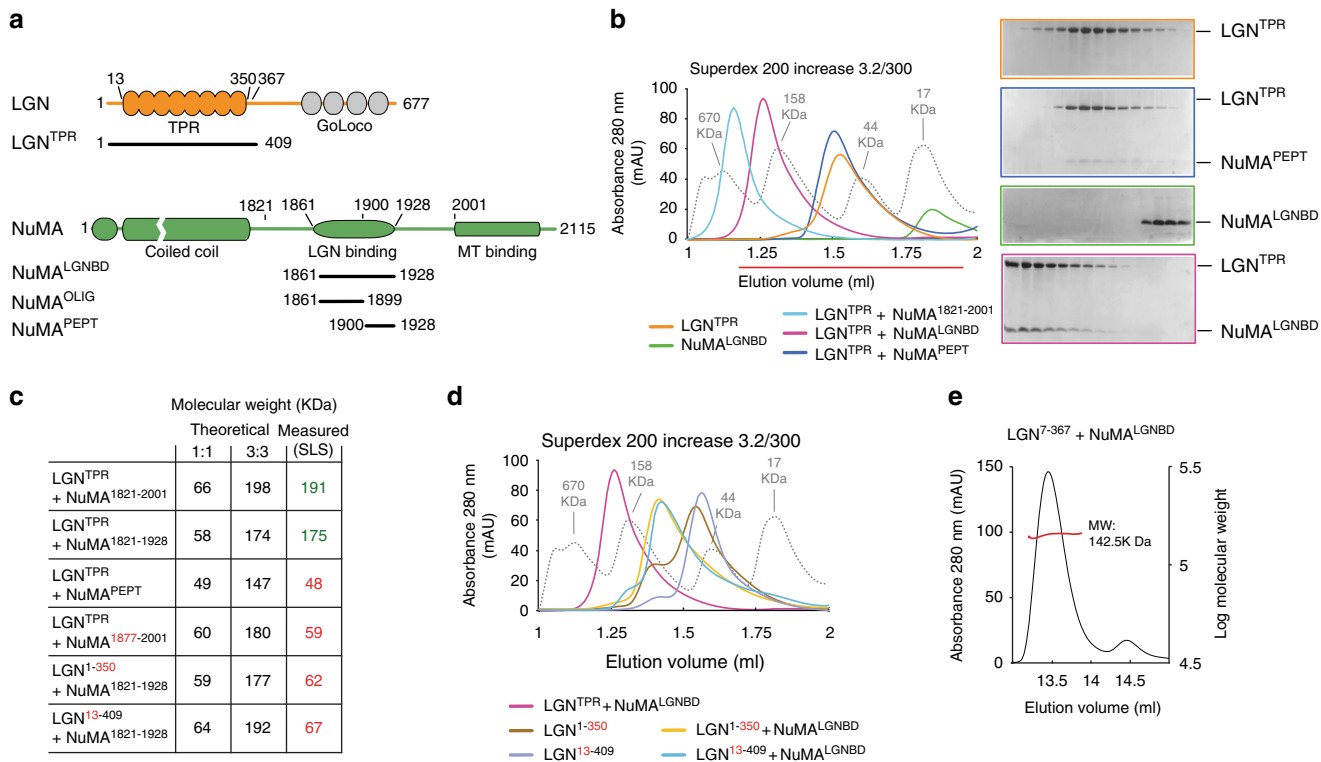

**Fig. 1** LGN and NuMA engage into high-molecular weight oligomers. **a** Cartoon representation of LGN and NuMA domain structures. Bold lines with numbers indicate protein subdomains being used in the in vitro binding assays. **b** SEC elution profiles of LGN$^{TPR}$ (20 μM) in complex with NuMA$^{1821-2001}$ (20 μM), NuMA$^{LGNBD}$ (20 μM) or NuMA$^{PEPT}$/NuMA$^{1900-1928}$ (40 μM). The run of globular molecular weight markers is indicated in gray. The early elution volume of LGN$^{TPR}$:NuMA$^{LGNBD}$ indicates they form higher molecular species compared with the 1:1 stoichiometric LGN$^{TPR}$:NuMA$^{PEPT}$/NuMA$^{1900-1928}$ dimer. Coomassie-stained SDS-PAGE in color-coded boxes show the protein composition of each elution profile. **c** Table of static light scattering (SLS) experiments conducted with the trimmed LGN:NuMA constructs used in Fig. 1b–d, and Supplementary Fig. 2a, b. SLS measurements shown in green indicate oligomerizing complexes, while red values belong to oligomerization-deficient LGN:NuMA pairs. **d** SEC analyses conducted with LGN$^{TPR}$ truncations and NuMA$^{LGNBD}$. The oligomerization determinants span residues 1–12 and 351–409 on LGN. **e** SLS analysis indicating that the LGN$^{7-367}$ and NuMA$^{LGNBD}$ engage in 142.5 kDa oligomers compatible with a 3:3 stoichiometry. This also suggests that LGN$^{7-367}$ is the minimal oligomerization domain of LGN

| Table 1 Data collection and refinement statistics | |
| --- | --- |
| | **LGN-NuMA** |
| Data collection | |
| Space group | P4₁2₁2 |
| Cell dimensions | |
| a, b, c (Å) | 153.94, 153.94, 732.95 |
| α, β, γ (°) | 90, 90, 90 |
| Resolution (Å) | 243.48-4.29 (4.40-4.29)[a] |
| $R_{sym}$ or $R_{merge}$ | 0.343 (7.028) |
| $I/\sigma I$ | 19.1 (2.1) |
| Completeness (%) | 99.9 (98.9) |
| Redundancy | 98.5 (88.7) |
| $CC_{1/2}$ | 0.998 (0.713) |
| Refinement | |
| Resolution (Å) | 141.90-4.31 |
| No. reflections | 60,923 |
| $R_{work}/R_{free}$ | 0.182/0.233 |
| No. atoms | |
| Protein | 37,562 |
| B-factors (Å²) | |
| Protein | 241.8 |
| R.m.s. deviations | |
| Bond lengths (Å) | 0.006 |
| Bond angles (°) | 1.21 |
| [a]Values in parentheses are for highest-resolution shell | |

central triangular cavity reflecting the threefold symmetry of the assembly (Fig. 2a–c). In such an arrangement, the flexible NuMA chains thread in-between two adjacent LGN subunits, and then line in the internal groove of the TPR domain (Fig. 2c, d). The interface between NuMA$^{1900-1928}$ and the TPR repeats of LGN in the hexamers is substantially identical to that observed in the crystallographic structure of LGN$^{15-350}$ in complex with the short NuMA fragment encompassing only residues 1900–1928[19]. The circular organization of the NuMA:LGN hetero-hexamers is allowed by the peculiar curvature of the TPR array of LGN induced by the longer helices of the TPR4, that are 10-residue longer than canonical TPR helices (see Supplementary Information in Culurgioni et al.[20]). Notably, the curvature of the LGN$^{TPR}$ superhelix in the LGN$^{TPR}$:NuMA$^{1861-1928}$ hexamers is more pronounced than that of LGN$^{TPR}$ in complex with NuMA$^{1900-1928}$ because of the more pronounced bending of TPR1–2 induced by the contacts between two adjacent TPR molecules in the donut (Supplementary Fig. 2).

The donut assembly is promoted primarily by the formation of a four-helix bundle containing the TPR8 and the capping helix of one LGN$^{TPR}$ protomer (named here LGN-1 for clarity), and the N-terminal helix preceding the TPR1 of the neighboring LGN$^{TPR}$ molecule (LGN-2), in a sort of molecular grip (Fig. 2b–d). This topology explains why deletion of residues 7–13 or 350–367 of LGN$^{TPR}$ impairs hexamer formation (Fig. 1c, d). In the hexamers, each NuMA molecule contacts two LGN$^{TPR}$: the C-terminal NuMA$^{1900-1928}$ portion binds to the inner groove of LGN$^{TPR}$,

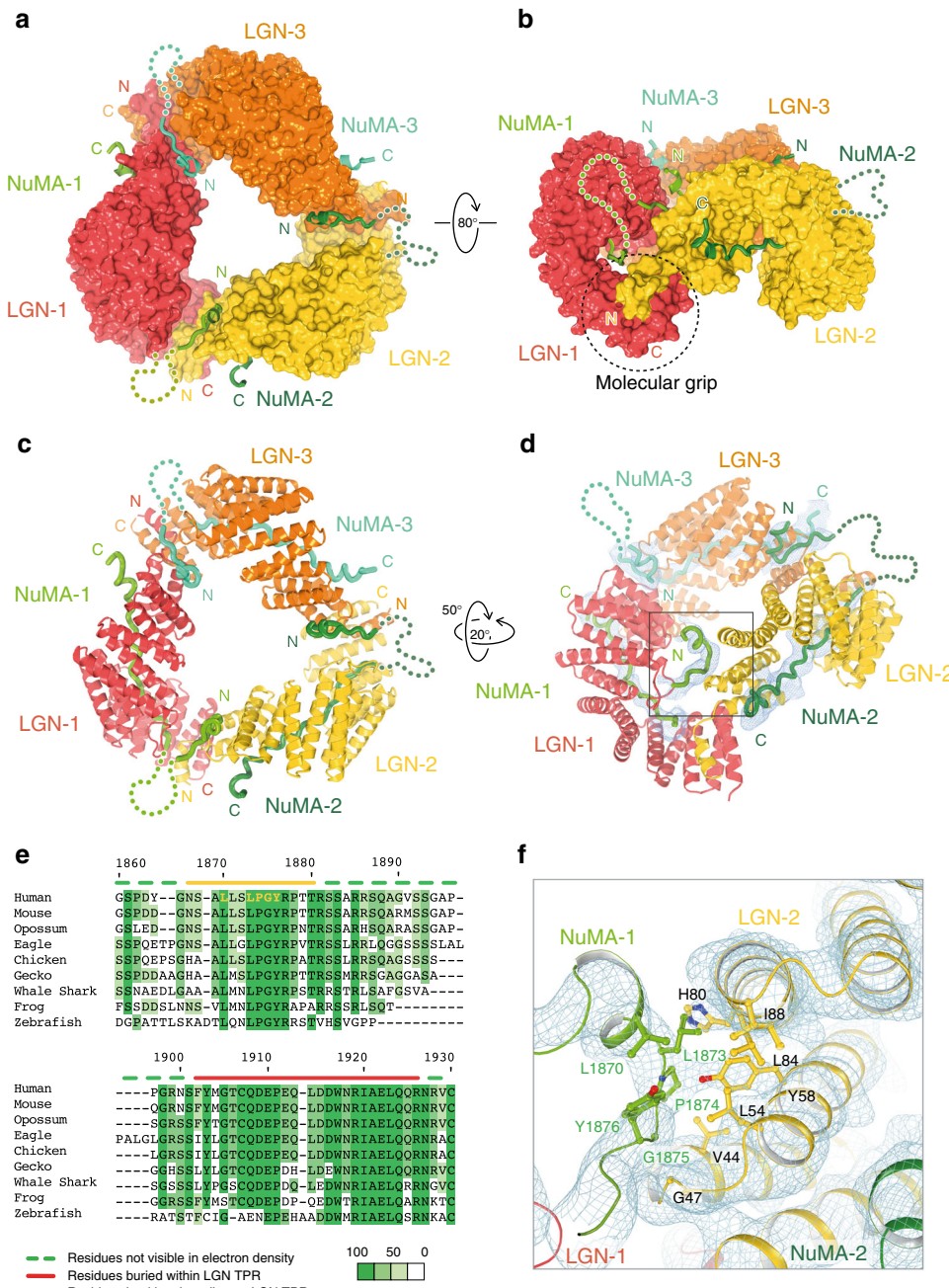

**Fig. 2** Structure of the hetero-hexameric LGN:NuMA complex. **a–b** LGN:NuMA hetero-hexamer with LGN molecules depicted with surface representations in red, orange and yellow, and NuMA molecules as coils in different shades of green. Unstructured regions are represented with dotted lines. Panel **b** highlights the 'molecular grip' between the N-terminal LGN-2 capping helix (yellow N-term) with the LGN-1 TPR8 (red C-term). **c–d** Cartoon representations of the hetero-hexameric complex. Panel **c** is in the same orientation of a, whilst panel **d** highlights the hooking mechanism of NuMA$^{LGNBD}$ N-terminal on the adjacent LGN$^{TPR}$. An electron density map (Fo-Fc omit map calculated omitting NuMA$^{LGNBD}$ chains) is displayed at 2.2-σ contour level. **e** Excerpt of NuMA sequence alignment encompassing the crystallized fragment, colored by percentage of sequence identity (details in Methods section). Residues built in the crystal structure are indicated with overlying yellow and red lines, those for which there is no electron density with overlying green dotted line. Residues of the NuMA hooking peptide contacting LGN are depicted in bold yellow. **f** Enlarged view of the region highlighted in the box of panel **d**, with ball-and-stick representation of the residues involved in the hooking interface between NuMA-1 and LGN-2. A 2Fo-Fc electron density map, contoured at 1-σ level, is displayed

while the N-terminal portion preceding NuMA residues 1900–1928 hooks on the outer surface of the adjacent LGN$^{TPR}$ to secure the toroidal molecular architecture (boxed in Fig. 2d). More specifically, residues 1864–1880 of NuMA-1 start with two helical turns interrupted at Pro1874, where the chain bends in an elongated polypeptide, and packs against helices-A/B of the second TPR motif of the LGN-2 subunit, facing Leu54$^{LGN-2}$ and

Tyr58$^{LGN-2}$ (Fig. 2f). Since binding of this initial NuMA fragment to the LGN$^{TPR}$ scaffold buries only about 400Å$^2$ of an accessible surface area, we asked whether its presence was essential for the hetero-hexamer formation. To this aim, we mutated to Ala the LGN$^{TPR}$ residues engaged in the NuMA$^{1861–1880}$ interaction, and tested the ability of the LGN$^{TPR}$-L54A-Y58A double mutant to form oligomers with NuMA$^{LGNBD}$. Analytical SEC experiments

revealed that disrupting the NuMA[1861–1880]:LGN[TPR] interface prevents hexamer formation (Supplementary Fig. 2), confirming that hooking of NuMA[1861–1880] on LGN[TPR] is essential for oligomerization. No electron density is visible for NuMA residues 1881–1897, it is observable again at the beginning of NuMA[1900–1928] that binds tightly to the inner concave groove of the TPR scaffold. Given the modest resolution of the data, to confirm the register of our tracing, we decided to generate a NuMA[LGNBD] construct lacking aa 1881–1897, and analyze by SEC the stoichiometry of the assemblies it formed with LGN[TPR]. We reasoned that if our map interpretation was correct, LGN[TPR]: NuMA[LGNBD-Δ1881–1897] complexes would be hetero-hexamers as the LGN[TPR]:NuMA[LGNBD] complexes. Conversely, if the visible electron density corresponded to the NuMA residues 1881–1897, we would observe a LGN[TPR]:NuMA[LGNBD-Δ1881–1897] complex with a 1:1 stoichiometry, eluting from SEC as the LGN[TPR]: NuMA[1877–1928] complex. When loaded on a Superdex-200 column, LGN[TPR] in complex with NuMA[LGNBD] or with NuMA[LGNBD-Δ1881–1897] eluted in the same fractions (Supplementary Fig. 2), fully supporting the molecular model that we had built. We conclude that NuMA has a bipartite LGN-binding domain consisting of a nanomolar affinity peptide spanning residues 1900–1928 and a disjunct upstream oligomerizing motif encompassing aa 1861–1880, which is essential for hetero-hexamer formation. Both stretches are evolutionarily conserved (Fig. 2e) and spaced by an intervening sequence of different lengths rich in Gly and Ser, as expected for an unstructured linker between functional binding elements. All interfaces within the hetero-hexamer are essentially identical, conferring a genuine threefold symmetry to the macromolecular arrangement. Collectively, our structural analyses revealed that LGN[TPR] engages with NuMA in hexameric rings, held together by contacts between the LGN helices flanking the TPR domain, i.e., the N-terminal helix and the capping helix, and residues 1861–1880 of NuMA preceding the NuMA[1900–1928] stretch.

**NuMA/LGN oligomerization is needed for planar divisions**. To promote spindle orientation, in metaphase NuMA is recruited to the cortex by direct interaction with LGN. To understand whether the multimeric nature of the LGN:NuMA assemblies is important for the activation of force generators, we set out to rescue spindle orientation defects caused by the loss of LGN by ectopic expression of an LGN construct unable to oligomerize, referred to as LGN-ΔOLIGO in the following paragraphs. Based on the molecular information provided by the structural analyses, LGN-ΔOLIGO lacks residues 1–12 (corresponding to the N-terminal helix) and 350–366 (corresponding to the capping helix), but retains its ability to bind NuMA with the TPR scaffold in a 1:1 stoichiometry (Fig. 3a). We first addressed the relevance of the NuMA:LGN oligomerization for the morphogenesis of Caco-2 cysts that grow as monolayered single-lumen spheres by oriented planar divisions, with the spindle aligned within the monolayer. To this aim, we generated Caco-2 cell lines ablated for LGN, and stably expressing LGN wild-type or LGN-ΔOLIGO at levels comparable with the endogenous protein (Supplementary Fig. 3). When plated in Matrigel, wild-type Caco-2 cells form single-lumen cysts by oriented division (Fig. 3b, top-left panel, Fig. 3c, d, and Supplementary Fig. 3). In contrast, Caco-2 cells lacking LGN undergo misoriented divisions and fail to organize a single lumen (Fig. 3b, top right, Fig. 3c, d). Ectopic expression of wild-type LGN in Caco-2 cells fully rescues the misorientation and multi-lumen phenotypes, while expression of LGN-ΔOLIGO does not (Fig. 3b bottom panels, and Fig. 3c–d, and Supplementary Fig. 3). Importantly, LGN loss does not impair apico-basal polarity[22]; thus, the defective cystogenesis observed upon LGN ablation can

be directly ascribed to misoriented mitoses. We conclude that NuMA:LGN oligomerization is essential for oriented planar division and correct cystogenesis.

We next set out to dissect the molecular mechanism underlying the misorientation phenotype observed in Caco-2 cysts using HeLa cells, that are more amenable to imaging, and when plated on fibronectin-coated coverslips, divide with the spindle parallel to the substratum[23]. Similarly to what we had done for Caco-2 cells, we first generated HeLa cell lines stably interfered for LGN and expressing LGN wild-type or the oligomerization-deficient mutant (Supplementary Fig. 3), and confirmed that also in this cellular system, the misorientation caused by LGN ablation is rescued by wild-type LGN but not by the oligomerization-deficient LGN-ΔOLIGO (Fig. 3e, f). To explain why LGN-ΔOLIGO cannot support spindle orientation, we first reasoned that NuMA:LGN oligomerization could favor cortical clustering of NuMA molecules, and hence of dynein/dynactin motors, this way effectively activating pulling forces on astral MTs. To test this hypothesis, we evaluated cortical levels of LGN, NuMA, and dynactin in the mitotic HeLa cell lines generated above. The current model for force generator assembly posits that Gαi moieties anchored at the plasma membrane recruit LGN, which in turn targets NuMA to the cortex to assemble dynein/dynactin[7]. Consistent with this model, quantification of LGN at the cortex showed that both LGN and LGN-ΔOLIGO enrich at the cortex to the same extent of the endogenous protein (Fig. 3g, h) because they all contain a proficient C-terminal GoLoco region. Cortical NuMA is lost upon LGN depletion (Fig. 3i, j, second left panel) but accumulates back in equal amounts upon re-expression of LGN wild-type or LGN-ΔOLIGO (Fig. 3i, j, last two right panels), indicating that NuMA is recruited to the cortex by the binary interaction with LGN without the need of oligomerizing. According to the notion that NuMA is the driver for force generator assembly, in the four analyzed HeLa cell lines, the distribution of the p150 subunit of dynactin and the light-intermediate chain 1 of dynein (LIC1) mirrors the behavior of NuMA (Fig. 3k, l, and Supplementary Fig. 3), because they are lost in LGN-ablated cells and present in LGN wild-type and LGN-ΔOLIGO HeLa cells. The LGN-ΔOLIGO construct cannot form high-order oligomers with NuMA, but is still able to bind NuMA in a 1:1 stoichiometry. To assess whether excess of LGN can overcome the requirement of oligomerization in generating cortical MT-pulling forces, we performed a spindle-rocking experiment[7] in which we overexpressed LGN wild-type or LGN-ΔOLIGO in HeLa cells stably expressing H2B-GFP, and filmed the metaphase plate oscillations (Supplementary Fig. 3). This analysis showed that large excess of LGN-ΔOLIGO triggers spindle rocking to the same extent of that observed upon overexpression of LGN-WT. This result is not unexpected, considering that the LGN constructs are massively overexpressed in transiently transfected HeLa cells, as compared to the endogenous protein (Supplementary Fig. 3), likely bypassing the regulatory mechanisms governing spindle positioning under physiological conditions. Whether NuMA:LGN oligomerization becomes dispensable upon cell treatments accumulating aberrantly LGN at the cortex[24] remains an open issue.

To corroborate these results obtained with LGN-ΔOLIGO, we exploited the information provided by the crystallographic structure of NuMA:LGN donuts to engineer also a NuMA mutant lacking residues 1861–1900 that still associates with LGN but does not form oligomers, that we refer to as NuMA-ΔOLIGO (Supplementary Fig. 4). We then used a HeLa cell line in which NuMA was stably ablated[9], and measured the spindle orientation angles of the cells plated on fibronectin-coated coverslides upon transfection of NuMA rescue constructs. Under these conditions, unperturbed cells aligned the spindle parallel to the substratum,

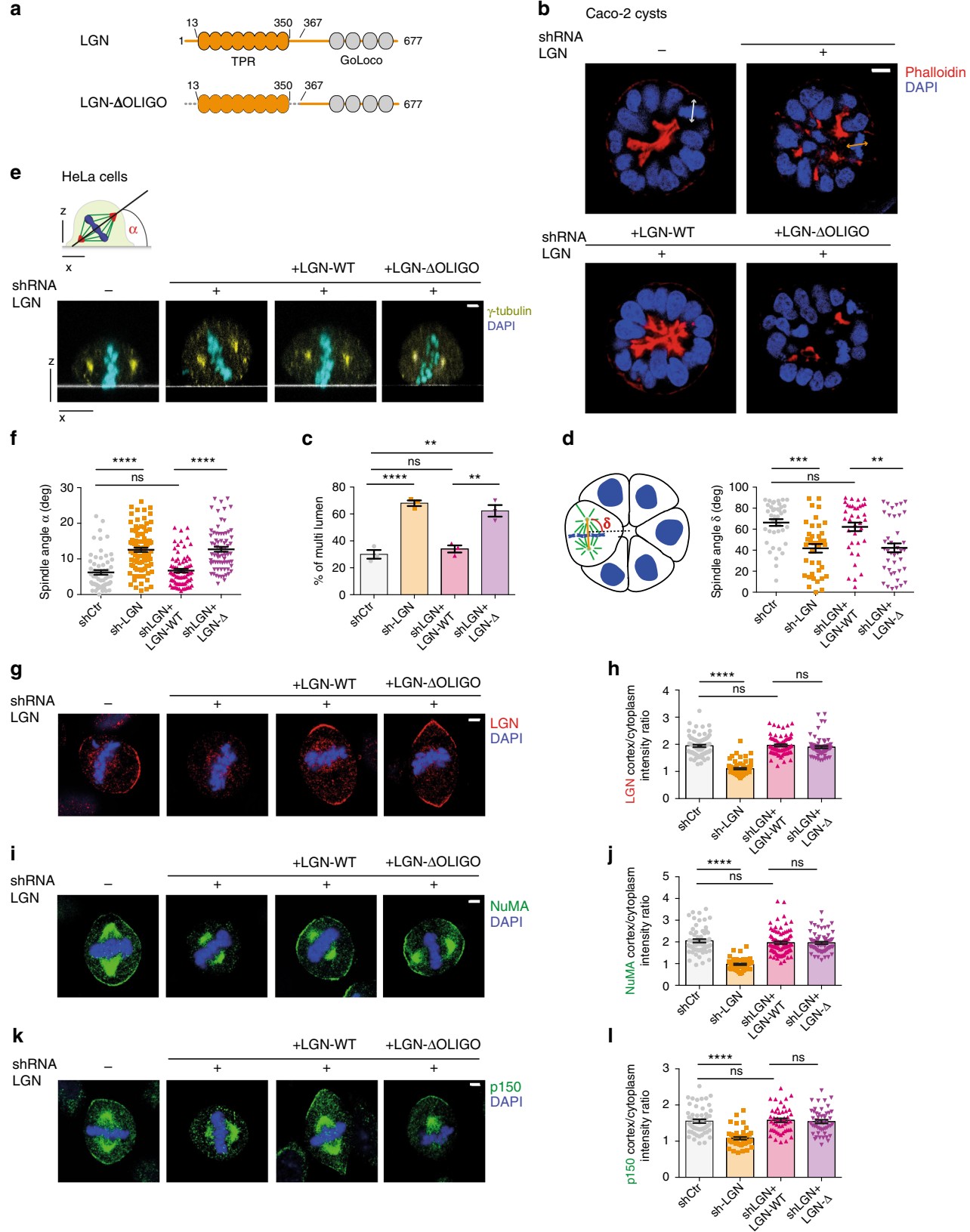

while cells lacking NuMA divide with a randomized orientation (Supplementary Fig. 4). Transient transfection of wild-type NuMA rescues misorientation defects, whereas transfection of NuMA-ΔOLIGO does not, similarly to what was observed with a NuMA mutant lacking the entire LGN-binding domain (Supplementary Fig. 4). Although these results are consistent with the ones obtained with LGN-ΔOLIGO, they cannot be fully ascribed to the cortical activities of NuMA in complex with LGN, because the LGN-binding domain of NuMA overlaps with a site (spanning residues 1944–2003 of human NuMA) involved in

**Fig. 3** Oligomerization-proficient LGN is required for mitotic spindle orientation. **a** Domain structure of LGN wild-type and oligomerization-deficient mutant. Dashed lines indicate fragments of LGN deleted in the rescue construct. **b** Confocal sections of Caco-2 cysts grown from cells wild-type (top left) or depleted of endogenous LGN (top right) and expressing C-terminal mCherry-tagged LGN wild-type (WT, bottom left) or oligomerization deficient LGN (LGN-ΔOLIGO, bottom right). Cysts were stained with phalloidin (red), and DAPI (blue). Lines indicate the mitotic spindle. **c** Quantification of cystogenesis of Caco-2 cells imaged in panel **b**. LGN-Δ stands for LGN-ΔOLIGO. Histograms show the percentage of multi-lumen cysts. Mean and SD are shown for 3 independent experiments, with $n > 60$. ***$p < 0.0001$; **$p < 0.01$; *$p < 0.05$ by Fisher's exact test. **d** Dot-plot with the distribution of metaphase spindle angle of Caco-2 cysts shown in Supplementary Fig. 3. Mean ± SEM are shown for 3 independent experiments, with $n > 36$. ***$p < 0.001$, ** indicates $p < 0.01$ by Krustal-Wallis test. **e** Confocal $x$–$z$ sections of HeLa cells depleted of endogenous LGN and expressing mCherry-tagged LGN-WT or LGN-ΔOLIGO. Cells were stained with γ-tubulin (yellow) and DAPI (cyan). Quantification of the orientation was performed by measuring the angle formed by a line passing through the spindle poles and the coverslip (white line). **f** Dot-plot with spindle angle distributions of HeLa cells imaged in panel **e**. Means ± SEM are shown for 4 independent experiments, with $n = 61$ for control, $n = 94$ for LGN-shRNA, $n = 78$ for LGN-depleted cells expressing LGN-WT, and $n = 74$ for LGN-depleted cells expressing LGN-ΔOLIGO. ****$p < 0.0001$ by Krustal–Wallis test. **g-i-k** IF of HeLa cells depleted of endogenous LGN and expressing mCherry-tagged LGN rescue constructs. Cells were stained for LGN (**g**), NuMA (**i**) or the dynactin subunit p150 (**k**). **h-j-l** Quantification of the cortical signals of HeLa cells in panels **g-i-k**, with histograms of the cortex-to-cytoplasm fluorescence ratio. Means ± SEM are shown for 3 independent experiments with $n > 50$. ****$p < 0.0001$ by Krustal-Wallis test. Scale bars, 5 μm for all HeLa cells, 10 μm for all Caco-2

spindle pole activities[25]. In line with these considerations, quantifications of the amounts of NuMA constructs at the spindle poles revealed that NuMA-ΔOLIGO and NuMA-ΔLGNBD accumulate significantly less at the poles than wild-type NuMA (Supplementary Fig. 4), implying that the inability of these constructs to rescue misorientation might be not only due to impaired oligomerization with LGN but also due to spindle assembly defects.

In summary, from the analyses in Caco-2 cysts and HeLa cells, we conclude that NuMA:LGN hetero-hexamers are fundamental for spindle positioning and epithelial morphogenesis promoted by planar cell divisions, although they are not required for targeting dynein/dynactin at the cortex.

**NuMA/LGN oligomers assemble in a multimeric protein network.** We next set out to understand how NuMA:LGN hetero-hexamers promote spindle orientation.

As a first step, we started testing whether the NuMA:LGN hexamers could form in cells. To assess the assembly of LGN molecules independently from the oligomerization driven by the dimerizing NuMA coiled-coil region, we generated a HEK293T cell line stably depleted of endogenous NuMA and expressing a C-terminal portion of NuMA encompassing the LGN-binding domain but not the coiled-coil (i.e., NuMA$^{1821-2115}$). We then co-transfected these cells with either GFP-LGN-WT and LGN-WT-FLAG or GFP-LGN-ΔOLIGO and LGN-ΔOLIGO-FLAG, and tested whether in mitotic lysates the GFP-tagged version of LGN could immunoprecipitate the FLAG-tagged version of LGN. Our experiment revealed that only LGN wild-type proteins immunoprecipitate each other together with NuMA$^{1821-2115}$, while LGN-ΔOLIGO cannot (Fig. 4a). This evidence supports the notion that LGN and NuMA can assemble higher-order oligomers in mitotic cells, independently of NuMA self-assembly, and that the same mutations impairing oligomerization in vitro disrupt LGN:NuMA oligomer formation in cells, indicating that the NuMA:LGN hetero-hexamers are key for multivalent mitotic NuMA:LGN interactions.

We next tested the oligomeric state of NuMA. First, we designed a NuMA construct encompassing residues 1592–1694, which are predicted to be the C-terminal portion of the NuMA coiled coil. Measurement of the molecular weight of purified NuMA$^{1592-1694}$ by static-light scattering showed that it forms homodimers (Fig. 4c). Importantly, co-immunoprecipitation of GFP-tagged and FLAG-tagged NuMA constructs from mitotic lysates confirmed that full-length NuMA proteins self-assemble in cells (Supplementary Fig. 4).

We then reasoned that the combination of full-length NuMA dimers with the 3:3 stoichiometry of the NuMA:LGN interaction would result in the formation of multimeric assemblies, in which the NuMA dimers are physically linked to different donuts. To test this idea, we assessed NuMA:LGN protein network formation using recombinant proteins. While the NuMA fragment 1592–2001 was insoluble, a chimeric NuMA construct in which the two functional stretches, i.e., the dimerizing coiled-coil (aa. 1592–1694) and the LGN-binding fragment (a.a 1821–2001), were connected by an artificial linker of eight Thr-Gly-Ser repeats (NuMA chimera henceforth, Fig. 4b), was soluble and could be purified to homogeneity. Upon incubation with LGN$^{TPR}$, NuMA chimera formed high-order oligomers eluting from a size-exclusion Superose-6 column at high molecular weight but well clear of the void volume (Fig. 4d). Static-light-scattering analyses confirmed that the NuMA-chimera:LGN$^{TPR}$ sample is a poly-disperse population of assemblies with a molecular weight upto 4.5 MDa (Fig. 4e). This result is fully consistent with our network hypothesis, predicting that a minimum of three NuMA dimers are needed to satisfy the 3:3 stoichiometry of the NuMA:LGN hetero-hexamers, but that larger multimeric clusters can form if the chains of each NuMA dimers engage with different donuts (Fig. 4f). When analogous SEC and SLS analyses were repeated with a sample assembled with NuMA chimera and LGN$^{TPR-ΔOLIGO}$, a significant shift toward a lower molecular weight was observed in the SEC elution profile (Fig. 4d), which was accompanied by a decrease in the molecular weight measured by SLS to about 136 kDa, corresponding to NuMA:LGN 2:2 complexes formed by the interaction of two LGN$^{TPR-ΔOLIGO}$ with the two copies of NuMA present in the NuMA-chimera dimer (Fig. 4g). Taken together, this evidence confirms that in vitro binding of dimeric NuMA moieties to LGN$^{TPR}$ results in high-order multivalent protein assemblies in which hetero-hexameric NuMA$^{LGNBD}$:LGN$^{TPR}$ donuts are connected to elongated coiled-coil regions of NuMA by flexible linkers.

**Cortical NuMA clusters dynein/dynactin with LGN.** We reasoned that if NuMA and LGN enable the formation of a multivalent protein network, it might be that it is precisely this supra-molecular organization of the microtubule motors to be key in sustaining the activation of pulling forces orienting the spindle. To test this idea, we designed an experimental setting capable of decoupling the cortical recruitment of NuMA from the assembly of oligomeric NuMA:LGN protein networks. To this aim, we took advantage of the notion that treatment of HeLa cells with the Aurora-A inhibitor MLN8237 causes massive accumulation of endogenous NuMA at the spindle poles accompanied by a misorientation phenotype, which can be bypassed by expression of a NuMA-LGN-GoLoco fusion protein (hereon NuMA-GoLoco) ectopically localizing

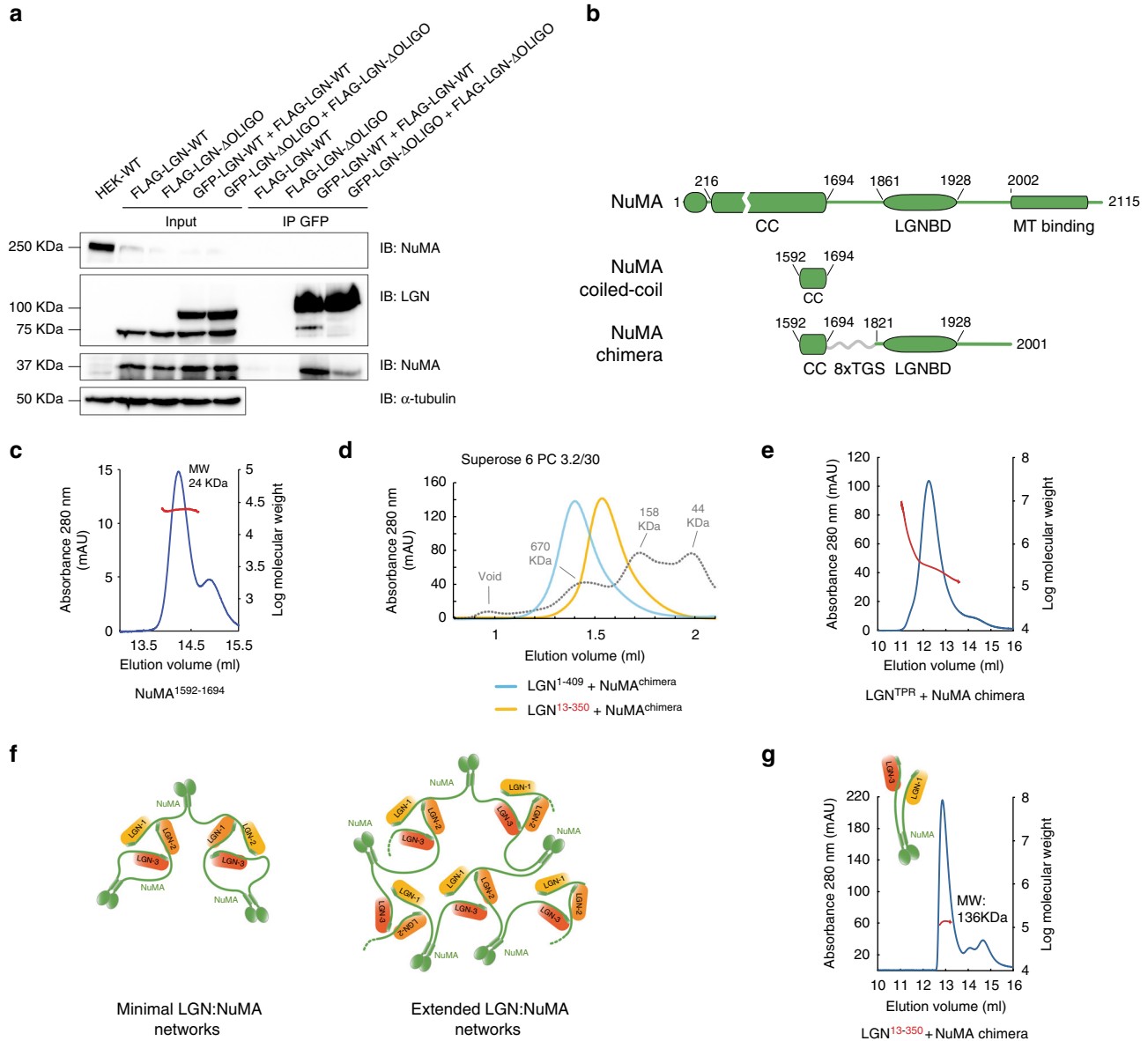

**Fig. 4** Binding of dimeric NuMA to LGN[TPR] generates protein networks. **a** NuMA:LGN form hetero-hexamers in mitotic cells. HEK293T cells stably depleted of endogenous NuMA and stably expressing NuMA[1821–2115] were transiently co-transfected with plasmids expressing human GFP-LGN wild-type and LGN-3xFLAG wild-type or GFP-LGN-ΔOLIGO and LGN-ΔOLIGO-3xFLAG. After 48 h, mitotic lysates were immunoprecipitated (IP) with anti-GFP antibodies conjugated to agarose beads, and immunoblotted with the indicated antibodies. Cells transfected with LGN-3xFLAG constructs were used as specificity control of the IP, and wild-type HEK293T cells were loaded to monitor NuMA depletion (IB anti-NuMA at a molecular weight of 250 kDa) and NuMA[1821–2115] expression (IB anti-NuMA at a molecular weight of 37 kDa). **b** Scheme of NuMA full-length, the C-terminal part of NuMA coiled-coil, and the NuMA chimera used to assemble NuMA:LGN[TPR] multimers in vitro. **c** SLS profile of NuMA[1592–1694] showing an average molecular mass of 24 kDa along the peak, as expected for a dimeric construct. **d** SEC elution profiles of the NuMA chimeric construct encompassing the coiled-coil and the LGNBD in complex with LGN[TPR] (LGN[1–409], blue trace) or LGN[TPR-ΔOLIGO] (LGN[13–350], yellow trace). Complexes of NuMA-chimera with the oligomerization-deficient LGN[TPR] construct elute later indicating that they have an average molecular weight lower than the ones assembled with LGN[TPR], as confirmed by the SLS analyses of Fig. 4e–g. The elution profile of globular molecular weight markers is shown in gray dotted lines. **e** SLS profile of the NuMA-chimera:LGN[TPR] sample indicating a polydispersed population of complexes with molecular masses in the range of 4.6 MDa to 130 kDa. **f** Scheme of different oligomers resulting from the assembly of NuMA with LGN[TPR]. The minimal stoichiometry of dimeric NuMA engaged in hetero-hexamers with LGN[TPR] is 6:6 (left). When each of the dimeric NuMA chains belongs to a different LGN[TPR] donut, higher order protein networks with diverse stoichiometry are possible (right). **g** SLS analysis of NuMA-chimera:LGN-TPR-ΔOLIGO showing that in the absence of NuMA:LGN oligomerization complexes with an average molecular mass of 136 kDa are formed, corresponding to 2:2 hetero-tetramers

at the cortex by direct binding to Gαi[9]. In this setting, we engineered an oligomerization-deficient GFP-NuMA-ΔOLIGO-GoLoco construct that localizes at the cortex but is unable to form high-stoichiometry NuMA:LGN assemblies (Fig. 5a). When probed in HeLa cells treated with MLN8237,

GFP-NuMA-ΔOLIGO-GoLoco did not rescue spindle misorientation in spite of accumulating at the cortex at the same levels of the orientation-proficient NuMA-GoLoco (Fig. 5b, c). This result indicates that it is not the presence, but the molecular organization of cortical NuMA in complex

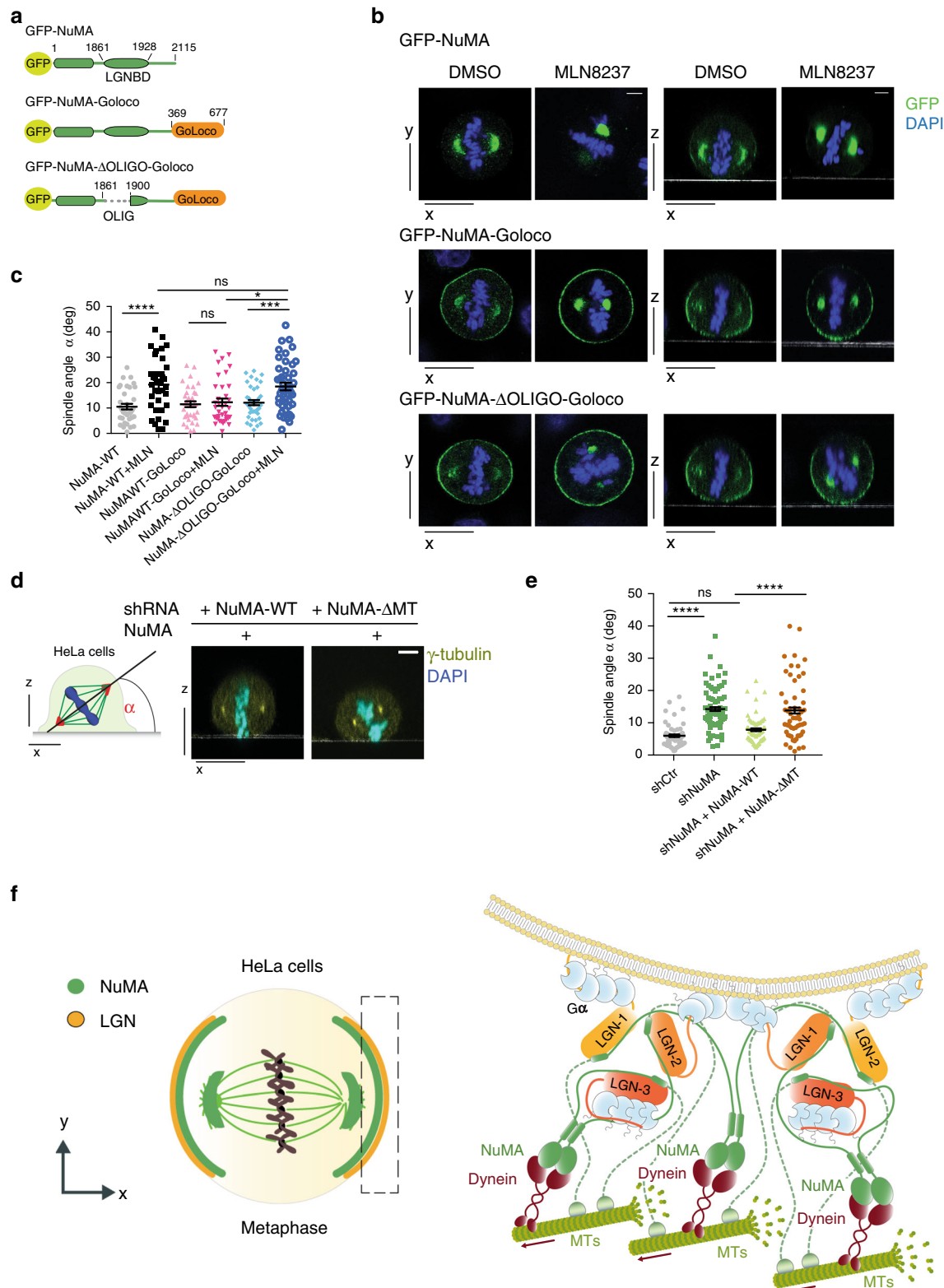

with LGN that is required for the onset of productive pulling forces.

Based on the current knowledge, spindle orientation is attained by traction forces exerted on astral microtubules by dynein/dynactin motors targeted to specialized cortical areas via NuMA: LGN proteins[7]. Intriguingly, the C-terminal region of NuMA spanning residues 2002–2115 contains a MT-binding domain compatible with concomitant binding to LGN[9], which is important for NuMA's ability to support pulling forces when ectopically localized at the cell cortex[8]. To start investigating the MT-binding activities of NuMA, we performed co-sedimentation assays with a purified C-terminal fragment encompassing the MT-binding domain (aa 2002–2115). NuMA$^{2002–2115}$ co-sediments with taxol-stabilized MTs, both in the presence and

**Fig. 5** NuMA:LGN cortical protein networks are essential to promote spindle orientation. **a** Schematic representation of GFP-NuMA constructs decoupling NuMA cortical recruitment and NuMA:LGN oligomerization. **b** Confocal x–y (left) or x–z (right) sections of mitotic HeLa cells expressing the indicated GFP-NuMA or GFP-NuMA-GoLoco constructs, treated with DMSO or MLN8237. Cells were stained with DAPI (blue). Scale bars, 5 μm. **c** Dot-plot illustrating the distribution of the mitotic spindle angles of metaphase HeLa cells shown in panel b. For each condition, means ± SEM are shown for three independent experiments, with n > 56. ****p < 0.0001; ***p < 0.001 and *p < 0.05 by Krustal-Wallis test. **d** Confocal x–z sections of metaphase HeLa cells depleted of endogenous NuMA and transfected with mCherry-NuMA wild-type or mCherry-NuMA-ΔMT. Cells were stained with γ-tubulin (yellow) and DAPI (cyan). Scale bars, 5 μm. **e** Dot-plot illustrating the distribution of the spindle axis angles for HeLa cells visualized in panel **d**, with control cell lines expressing a scrambled shRNA or the shRNA-NuMA used to deplete the endogenous protein. For each condition, means ± SEM are shown for four independent experiments, with n > 55. ****p < 0.0001 by Krustal-Wallis test. **f** Model of the cortical dynein/dynactin:NuMA:LGN:Gαi protein network pulling on astral MTs based on our studies

in the absence of tubulin tails (Supplementary Fig. 5), indicating that NuMA recognizes directly the MT lattice. We next tested the ability of NuMA to interact with tubulin dimers. SEC experiments confirmed that NuMA$^{2002–2115}$ associates stoichiometrically with tubulin dimers (Supplementary Fig. 5). This evidence suggests that cortically anchored NuMA targets cortically dynein/dynactin motors, possibly regulating the dynamics of astral MT plus ends via its MT-binding domain. This way, NuMA could facilitate astral MT shrinkage while dynein moves toward the minus end.

Last, to address the relevance of the MT binding of NuMA for orientation, we generated a NuMA-ΔMT mutant truncated at residue 2001 (Supplementary Fig. 5) and analyzed its spindle orientation properties in HeLa cells. Contrary to wild-type NuMA, NuMA-ΔMT cannot revert the misorientation phenotype of HeLa cells lacking endogenous NuMA (Fig. 5d, e). This result can be ascribed to the evidence that NuMA-ΔMT does not enrich at the cortex (Supplementary Fig. 5). However, we suspected that binding of NuMA to MT could also be implicated in spindle pole organization, this way indirectly contributing to correct spindle orientation. To test this possibility, we compared the amounts of mCherry-NuMA wild-type and NuMA-ΔMT that accumulate at the spindle poles in mitotic HeLa cells. For these experiments, we used HeLa cells depleted of the endogenous protein in order to prevent homodimerization. As expected, mCherry-NuMA-ΔMT cannot decorate spindle MTs emanating from the poles, but is retained only on centrosomes, and overall accumulates at the poles at lower levels than the wild-type counterpart (Supplementary Fig. 5). We conclude that the microtubule-binding domain of NuMA plays essential roles at the cortex and at the spindle poles, and that these activities are essential to grant proper spindle orientation.

## Discussion

Spatial organization of traction forces pulling on astral microtubules to orient the spindle is achieved by recruitment of cytoplasmic dynein/dynactin at the cortex. Here, we show that the dynein adaptor NuMA and membrane-localized LGN assemble in oligomers that can form multivalent interactions key to sustain effective pulling on astral MTs. Biochemical and structural studies revealed that these NuMA:LGN networks are organized on hetero-hexameric modules in which three TPR domains of LGN interact with the C-terminal LGN-binding stretch of dimeric NuMA (the NuMA$^{LGNBD}$) in a donut-shaped architecture. Spindle orientation assays conducted in Caco-2- polarized three-dimensional cysts and HeLa cells in adhesion, indicated that oligomerization of NuMA with LGN is essential for spindle orientation processes and epithelial morphogenesis, although it does not mediate cortical recruitment of dynein/dynactin. Finally, we found that the MT-binding domain of NuMA residing in the 2002–2115 C-terminal portion is also implicated in spindle assembly and orientation. Based on the evidence that NuMA recognizes the MT lattice, as well as tubulin dimers, we

speculate that NuMA contributes to set the dynamic rate at the plus ends and to increase dynein processivity on the MT lattice (Fig. 5f).

The major findings of the current study stem from the structural evidence that LGN$^{TPR}$ and NuMA$^{LGNBD}$ form hetero-hexameric rings whose determinants are the LGN helices preceding and following the eight TPR repeats, and a NuMA motif spanning residues 1861–1899 preceding the high-affinity LGN-binding peptide. The toroidal architecture of the hexamers is allowed by the characteristic TPR array of LGN featuring a long TPR4 that confers a pronounced curvature to the scaffold. In addition, hooking of the N-terminal helix of one LGN molecule onto the last TPR repeat of the subsequent one in the donut generates a molecular tension, resulting in an increased curvature of the TPR scaffold (Supplementary Fig. 2). Interestingly, we observed that the TPR domain of LGN reveals a great conformational versatility, which allows the assembly of oligomers of different geometry with the diverse LGN-binding partners, as visible in the LGN:Insc assembly, where the LGN$^{TPR}$ domains engage in a head-to-head tetrameric interaction with Insc[26].

In spite of the overall conservation of the NuMA/LGN pathway throughout species, it is not trivial to predict from sequence comparison if all orthologs of human LGN and NuMA can form hexamers with a similar architecture. For instance, Drosophila Pins have a 40-residue long N-terminal sequence before the TPR domain with poor propensity to adopt a helical conformation, and the Pins-binding domain of Mud (the counterpart of NuMA in flies) is not sufficiently well-defined to allow an accurate prediction of the oligomerizing properties of Mud.

An important implication of the current NuMA:LGN structural characterization is that the combination of NuMA homodimers with NuMA:LGN hetero-hexamers may foster the assembly of a subcortical protein network that clusters dynein/dynactin. The evidence that in HeLa cells, oligomerization-deficient LGN and NuMA constructs can recruit correct amounts of dynein/dynactin at the cortex but cannot sustain spindle orientation, supports the notion that the multimeric nature of the NuMA:LGN protein network is key in activating the motor activity of cortical dynein. This is an unexpected result, highlighting how force generators rely on the self-organization of large localized protein assemblies that are ultimately instructed by restricted Gαi-GDP pools triggering LGN cortical recruitment. The fact that LGN binds four Gαi-GDP subunits in a cooperative manner confers to the pathway the ability of responding quickly to an initial cortical Gαi-GDP cue that ignites the formation of NuMA:LGN complexes. In some respect, the overall activation of cortical force generators is reminiscent of the activation of the TNF receptor Fas, that upon extracellular ligand binding, oligomerizes triggering the assembly of an intracellular Fas-FADD proteinaceous platform that is essential for apoptotic signaling[27]. In the case of the Gαi-GDP, it will be interesting to explore if in vertebrate cells, specialized GPCRs are responsible for creating an initial pool of Gαi-GDP starting force generators' activation, as it was demonstrated for Drosophila neuroblasts[28].

Elegant optogenetic reconstitution of dynein/NuMA/LGN pathway in HeLa cells revealed that targeting of dynein at the cortex is not sufficient to ensue pulling forces positioning the spindle, while cortical recruitment of dynein by ectopic targeting of NuMA results in the formation of dotted patterns that are required to promote effective spindle pulling[8]. These clusters seem to depend on a conserved and hydrophobic motif of NuMA positioned at residues 1768–1777, which belong to the linker region between the coiled-coil and the LGNBD, and we know makes the recombinant protein unstable in vitro. Although it is possible that these NuMA cortical clusters are affected by the ectopic targeting system, their appearance is consistent with the requirement of a supramolecular organization of force generators. Our result demonstrates that cortical multivalent interactions mediated by NuMA:LGN hexamers  are key in triggering the formation of a protein network that sustains spindle placement. Whether the clustering reported by Okumura and colleagues relies on a mechanisms that complements the NuMA:LGN hetero-hexamers of the current study, and can therefore synergize with them to support a robust spindle pulling, will need to be explored further.

NuMA is the dynein/dynactin adaptor that in mitosis assists spindle assembly and orientation processes[7,29]. We recently discovered that the last hundred residues of NuMA code for a MT-binding region that associate directly to MTs, whose function is to date poorly understood[9]. We discovered that in vitro NuMA$^{2002-2115}$ co-sediments with taxol-stabilized MTs regardless of the presence of the negatively-charged tails, suggesting that it recognizes the MT lattice. Interestingly, the same fragment enters a 1:1 complex with tubulin dimers, indicating that NuMA also interacts with depolymerized tubulin (Supplementary Fig. 5). Collectively, this evidence is consistent with a role of NuMA in regulating astral microtubule plus ends dynamics while the dynein/dynactin/NuMA complex slides towards the spindle pole, possibly conferring processivity to the motor. The findings that in HeLa cells a NuMA truncation mutant lacking the MT-binding region cannot rescue misorientation defects fully supports this notion, although more data are needed to uncouple the spindle orientation and spindle assembly activity of dynein/dynactin/NuMA.

In conclusion, our studies uncovered the existence of NuMA$^{LGNBD}$:LGN$^{TPR}$ hetero-hexamers, which can form multimeric networks of LGN/NuMA complexes at the cortex, whose assembly is likely triggered by localized pool of Gαi-GDP molecules. Such protein complexes favor dynein/dynactin cortical clusters, and are essential for the spatial organization of dynein-dependent pulling forces positioning the spindle in HeLa cells and in polarized epithelial cysts. How the Gαi GDP/GTP cycle, and the numerous NuMA interactors and post-translational modifications affect the stoichiometry and the spatial arrangement of the NuMA/LGN complexes remains to be addressed in vitro and in vivo.

## Methods

**Protein expression and purification**. GST-LGN$^{1-350}$, GST-LGN$^{13-409}$, GST-LGN$^{1-409}$ (GST-LGN$^{TPR}$ in the text), GST-LGN$^{7-367}$, and GST-NuMA$^{1861-1928}$ (NuMA$^{LGNBD}$ in the text) were cloned into pGEX-6P1 vector (GE Healthcare) and expressed in BL21 Rosetta E. coli cells (Novagen) as indicated in Carminati et al.[30]. NuMA$^{LGNBD-Δ1881-1897}$ was generated by substitution of residues 1881–1897 of NuMA with a Thr-Gly-Ser triplet on the GST-NuMA$^{1861-1928}$ vector using the QuikChange mutagenesis kit (Agilent). Cells were lysed in 0.1 M Tris-HCl pH 8.0, 0.3 M NaCl, 10% glycerol, 0.5 mM EDTA, and 1 mM DTT, and cleared for 1 h at $100,000 \times g$. Proteins were first affinity purified on glutathione beads (GSH), and then incubated with PreScission protease (GE Healthcare) overnight at 4 °C to remove the GST-tag. Cleaved LGN constructs were eluted from the GSH beads in a desalting buffer consisting of 20 mM Tris-HCl pH 8.0, 40 mM NaCl, 5% glycerol, 1 mM DTT, and loaded on a 6-ml Resource-Q ion-exchange column. Bound proteins were eluted by a salt gradient from 40 mM to 450 mM NaCl over 20 column volumes. To remove chaperone contaminants from LGN$^{7-367}$, after the ion-

exchange column the protein was incubated on ice for 1 h with 1.5 mM ATP supplemented with 1.5 mM MgCl$_2$, and further polished on a Superdex-200 column equilibrated in a buffer containing 20 mM Tris-HCl pH 8.0, 0.15 M NaCl, 5% glycerol, and 1 mM DTT. NuMA$^{LGNBD}$ was gel filtered in the same buffer on Superdex-200 right after PreScission tag removal. For crystallization experiments, LGN$^{7-367}$ and NuMA$^{LGNBD}$ were combined in a 1:1.3 molar ratio, and the resulting complex was purified on a Superdex-200 column equilibrated in 10 mM Tris-HCl pH 8.0, 0.15 M NaCl, and 1 mM DTT. Peak fractions were analyzed by SDS-PAGE analysis, pooled and concentrated to 14 mg/ml prior freezing at −80 °C.

The NuMA C-terminal fragment spanning residues 2002–2115 used in the MT co-sedimentation assays and SEC analysis of Supplementary Fig. 5, and NuMA$^{1592-1694}$ produced for SLS analysis, were cloned into a pETM14 vector (Novagen), expressed in BL21 E. coli cells by overnight induction with 0.2 mM IPTG at 20 °C, and purified by affinity and cation exchange chromatography as previously described[9]. For the MT co-sedimentation assay of Supplementary Fig. 5, Ndc80$^{Bonsai}$ was purified as reported by Ciferri et al.[31].

The chimeric construct of NuMA (NuMA-chimera in the text) was generated as follows: NuMA residues 1592–1694 were fused to residues 1821–2001 by means of an artificial linker consisting of 8 Thr-Gly-Ser (TGS) repeats. The length of the linker was chosen based on the crystallographic structure to allow the formation of LGN$^{TPR}$:NuMA-chimera protein networks in which each chain of the dimeric NuMA-chimera could enter a complex with two diverse NuMA:LGN hetero-hexamers. The chimera was built starting from a NuMA construct spanning residues 1592–2001 cloned into a pETM14 vector (Novagen). Deletion of the 1695–1820 NuMA region coupled to the insertion of the linker was achieved by PCR amplification of the pETM14-NuMA$^{1592-2001}$ template by 5′-phosphorylated primers, each of which harbored a complementary sequence to the NuMA regions being joined and an overhang sequence coding for 4 TGS triplets. The amplified product was then digested with DpnI for 1 h at 37 °C, cleaned by a PCR purification kit (Qiagen), and ligated by T4 ligase. NuMA-chimera was expressed in BL21 Rosetta E. coli cells (Novagen) by 5 h induction with 0.5 mM IPTG at 20 °C. Cells were lysed in 0.1 M Tris-HCl pH 8.0, 0.3 M NaCl, 5% glycerol, 5 mM imidazole, and 1 mM DTT, and cleared for 1 h at 100,000×g. Clear lysates were injected on a HiTrap chelating column (GE Healthcare) loaded with Ni$^{2+}$, and NuMA-chimera was eluted with a 5 mM–0.25 M imidazole gradient. Eluted fractions were dialyzed overnight at 4 °C against a buffer containing 20 mM Tris-HCl pH 8.0, 40 mM NaCl, 5% glycerol, 1 mM DTT, while incubating with PreScission protease (GE Healthcare) to remove the histidine-tag. The protein was then injected onto a Resource-Q anion exchange column, and eluted with a gradient of 40 mM−0.35 M NaCl in 20 column volumes. NuMA-chimera was further purified on a Superose-6 column equilibrated in 20 mM Tris-HCl pH 8.0, 0.15 M NaCl, 5% glycerol, and 1 mM DTT. Eluted fractions were pooled, concentrated, and frozen at −80 °C. To isolate samples of oligomeric NuMA:LGN complexes suitable for Static-Light-Scattering analysis, LGN$^{TPR}$ or LGN$^{TPR-ΔOLIGO}$ were combined with NuMA-chimera in a 1:1.2 molar ratio and separated on a Superose-6 column.

**Analytical size exclusion chromatography (SEC)**. For SEC analyses of Fig. 1 and Supplementary Fig. 2, LGN and NuMA variants were mixed in equimolar amounts (20 µM), loaded on a Superdex-200 Increase 3.2/300 column (GE Healthcare) equilibrated in 20 mM Tris-HCl pH 8.0, 0.150 M NaCl, 5% glycerol, and 1 mM DTT, and eluted in 50 µl fractions. LGN$^{TPR}$:NuMA$^{PEPT}$ were combined in a 1:2 molar ratio. Eluted species were monitored by absorbance at 280 nm and subsequently checked by SDS-PAGE followed by Coomassie staining.

SEC analysis of Supplementary Fig. 5 were conducted loading a complex assembled with 26 µM of NuMA$^{2002-2115}$ and 20 µM αβ-tubulin hetero-dimers on a Superdex-200 Increase 3.2/300 column equilibrated in GT buffer (80 mM PIPES pH 6.8, 1 mM MgCl$_2$, 1 mM EGTA) supplemented with 60 mM NaCl and 1 mM DTT. Eluted species were analyzed by Tris-Tricine-SDS-PAGE and visualized by Coomassie staining.

**Static-Light-Scattering measurements**. Static-Light-Scattering (SLS) analyses of Figs. 1c–e and 4c–g were performed on a Viscotek GPCmax/TDA instrument equipped with two TSKgel G3000PWxl columns (Tosoh bioscience) in series. Typically, 75 µl of purified samples concentrated at about 1.5–2 mg/ml were loaded on the columns.

**Crystallization and structure determination**. The LGN$^{7-368}$:NuMA$^{LGNBD}$ complex at 14 mg/ml was supplemented with 20 mM TCEP and screened for crystallization using commercially available screen kits in a 1:1 volume ratio. Crystallization experiments were conducted in 200 nl vapor diffusion sitting drops with a Cartesian Honeybee nanodispenser (Genomic Solutions) in three-square-well CrystalQuick Greiner plates. Diffraction-quality crystals were obtained using the Molecular Dimensions Ltd PACT screen at 20 °C at half concentration, with a reservoir containing 10% PEG3350, 0.05 M Bis-Tris propane pH 7.5, and 0.1 M Na-formate or 0.1 M Na-acetate trihydrate. For data collection, crystals were transferred to a cryo-buffer (reservoir buffer supplemented with 25% glycerol) and flash-frozen in liquid nitrogen. X-ray diffraction data were collected to 4.2–5.0 Å resolution at I04 and I04–1 beamlines at Diamond Light Source, Didcot, United

Kingdom (visits nt5966 and nt5967, respectively), exploiting the kappa-goniometer reorientation in order to optimize data acquisition along the $c$ axes. All data were initially processed with XDS implemented in xia2[32] to define the crystallographic space group, unit cell and data collection statistics. Thirteen datasets were selected for merging according to their data quality (higher resolution limits, completeness, and lower $CC_{1/2}$) and to the degree of crystal isomorphism. Combination and data merging were carried out with the help of the BLEND[33] computer program. After the first run in analysis mode only eleven datasets were selected to be further combined into a single and final dataset resolution limit of 4.3 Å (combination mode). Indeed, the removal of two datasets caused a significant improvement in crystal isomorphism (Linear Cell Variation drop from 392.15 Å to 1.61 Å), while additional removal of individual diffraction images caused a substantial elimination of intensities affected by radiation damage. The merged dataset was used for molecular replacement using a search model of LGN$^{TPR}$:NuMA$^{1900-1928}$ obtained by aligning and combining chain A (LGN$^{TPR}$) and chain B (NuMA$^{1900-1928}$) of pdb entries 3SF4 and 3RO2, respectively. Molecular replacement was performed with Phaser[34], which automatically found eight copies of LGN$^{TPR}$:NuMA$^{1900-1928}$ dimer. After placing some helices manually in the clearest electron densities, the position of the remaining four LGN$^{TPR}$:NuMA$^{1900-1928}$ dimers became evident enough to place them manually into the densities. The model was progressively optimized by iterative cycles of low resolution *jelly body* refinement in Refmac[35] and manual model building in Coot[36]. Additional steps of refinement were carried out with LOw REsolution STructure Refinement (LORESTR) in the ccp4 suite and phenix.refine in the Phenix suite[37], making use of Feature Enhanced Maps. The final model is refined to $R_{work}/R_{free}$ values of 0.182/0.233, and contains 4 copies of LGN$^{TPR}$ and of NuMA$^{LGNBD}$ hetero-hexamers in the asymmetric unit. PyMOL was used to generate all the illustrations of the structure (http://www.pymol.org).

**Sequence alignment**. NuMA sequences from *Homo sapiens* (Uniprot entry Q14980), *Mus musculus* (Uniprot entry E9Q7G0), *Monodelphis domestica* (Uniprot entry F7ELR7), *Rhincodon typus* (NCBI entry XP_020391007), *Danio rerio* (NCBI entry NP_001316910), *Gallus gallus* (NCBI entry NP_001177854), *Gekko japonicas* (NCBI entry XP_015270477), *Xenopus laevis* (NCBI entry XP_018103292), and *Haliaeetus leucocephalus* (NCBI entry XP_010574983) were aligned with MUSCLE[38] and colored by percentage of identity in Jalview[39].

**MT co-sedimentation assays**. Tubulin (Cytoskeleton Inc.) was polymerized into stable microtubules according to the producer's instructions. Microtubule co-sedimentation assays of Supplementary Fig. 5 were carried out as in Gallini et al.[31]. Briefly, microtubules were diluted to a final concentration of 9 μM in general tubulin (GT) buffer (80 mM PIPES pH 6.8, 1 mM MgCl$_2$, 1 mM EGTA) supplemented with 1 mM GTP, 50 μM Paclitaxel and 60 mM NaCl. In order to remove the C-terminal tubulin tails, microtubules were treated with 200 μg/ml subtilisin A (Sigma-Aldrich) for 30 min at 30 °C. Proteolysis was stopped with the addition of 10 mM PMSF. Microtubules with and without tails were incubated for 10 min at RT with 5 μM NuMA$^{2002-2115}$ or 1 μM Ndc80$^{Bonsai}$ [31], in a final volume of 50 μl. Reactions were transferred onto 100 μl of cushion buffer (80 mM PIPES pH 6.8, 1 mM MgCl2, 1 mM EGTA, 50 μM Paclitaxel, 50% glycerol) and ultracentrifuged for 15 min at 400,000 × g at 25 °C in a Beckman TLA100 rotor. Pellets and supernatants were analyzed by SDS-PAGE and visualized by Coomassie staining.

**Cell culture**. HeLa cells (ATCC, CCL-2) and HEK293T (ATCC, CRL-11268) cells were cultured at 37 °C in a 5% CO$_2$ atmosphere, in Dulbecco's Modified Eagle Medium (DMEM) supplemented with 10% FBS, 1% L-glutamine and antibiotics. For all the experiments, HeLa cells were plated on fibronectin-coated coverslips (5 μg/ml, Roche) and pre-synchronized with a single thymidine block/release. Briefly, cells were treated with thymidine (2.5 mM, Sigma T1895) for 24 h, and then fixed 8 h after the release. Caco-2 cells were cultured at 37 °C in a 5% CO$_2$ atmosphere, in DMEM supplemented with 20% FBS, 1% L-glutamine, 1% NaHCO$_3$, 1% non-essential amino acid, and antibiotics. To inhibit Aurora-A, HeLa cells were pre-synchronized by thymidine arrest/release and 50 nM MLN8237 (Selleck Chemicals) were added to the medium 6 h after release[9]. Cells were fixed after 9–10 h from release.

To produce cysts, Caco-2 single cells (ATCC, HTB-37) were plated either on top of matrigel (for multilumen formation analysis) or matrix-embedded (for spindle angle measurement). For multilumen analysis, cells were resuspended in medium supplemented with 2.5% matrigel to a final concentration of 30,000 cells/ml, and then plated in matrigel-precoated eight-well chamber slides (Ibidi). At day 5, cells were treated with 0.1 μg/ml Cholera toxin for 16 h and then fixed. For the spindle orientation experiment, cells were resuspended to a final concentration of 60,000 cells/ml in medium 40% matrigel, and 100 μl of the suspension was plated for each well of the eight-well chamber. At day 2, cells were treated with 10 μM RO-3306 (Sigma SML0569) for 16 h, and then fixed 45 min after the treatment.

To deplete LGN expression, the shRNA sequence GGATGTAGTGGGA AACAAT was cloned into a pll3.7 lentiviral vector carrying a GFP reporter, and used to generate stably interfered HeLa and Caco-2 cell lines. Protein depletion was monitored by western blot and immunofluorescence. For the knockdown of NuMA, a lentiviral vector carrying a GFP reporter and puromycin resistance, and expressing the NuMA shRNA CAUUAUGAUGCCAAGAAGCAGCAGA ACCA[30] was used to generate stably interfered HeLa (ATCC, CCL-2) and

HEK293T (ATCC, CRL-11268) cell lines. To rescue the misorientation phenotype of LGN-depleted HeLa and Caco-2 cells, an sh-resistant C-terminally tagged LGN-mCherry construct was cloned into a pCDH lentiviral vector under the Ubc promoter (SBI System Bioscience). To obtain the oligomerization-deficient LGN construct (LGN-ΔOLIGO), the LGN-mCherry gene was further engineered to remove the coding sequence of residues 1–12 and 350–367. The pCDH lentiviral vectors obtained this way were used to generate stable HeLa and Caco-2 cell lines.

To perform the immunoprecipitation experiment, the C-terminal fragment of NuMA, encompassing residues 1821–2115, was cloned into a pCDH lentiviral vector, and used to generate a HEK293T cell line knockdown for endogenous NuMA and expressing NuMA$^{1821-2115}$. To rescue misorientation of NuMA-ablated HeLa cells, an sh-resistant pCDH-mCherry NuMA vector previously generated was used[9]. To obtain the oligomerization-deficient NuMA (NuMA-ΔOLIGO) and the NuMA mutant unable to bind LGN (NuMA-ΔLGNBD), the mCherry-NuMA gene was engineered to remove either the residues 1861–1900 or 1861–1928. To produce a NuMA construct unable to bind microtubules, a stop codon at residue 2002 was introduced into pCDH-mCherry-NuMA full-length by QuikChange mutagenesis (Agilent). To rescue the orientation in cells treated with MLN8237, a pCDH lentiviral vector containing the fusion protein GFP-NuMA-GoLoco protein previously generated was used[9]. The NuMA-ΔOLIGO mutant was then subcloned in the pCDH-GFP-GoLoco vector. All the pCDH vectors were transfected into HeLa cells using Lipofectamine 3000 (Invitrogen) according to manufacturer's instruction.

**Immunoprecipitation experiments**. For experiment in Fig. 4a, LGN-WT and LGN-ΔOLIGO were cloned in pEGFP-C1 (Clonetech). The same constructs were cloned also into pCDH with an engineered C-terminal 3xFLAG tag. HEK293T cells lacking endogenous NuMA and stably expressing NuMA$^{1821-2115}$, described in the previous section, were co-transfected with 0.25 μg of pEGFP-LGN-WT and 10 μg of pCDH-LGN-WT-3xFLAG, or 0.25 μg of pEGFP-LGN-ΔOLIGO and 10 μg of pCDH-LGN-ΔOLIGO3xFLAG. After 48 h from transfection, cells were treated with 0.33 μM nocodazole for 16 h. Mitotic cell lysates were prepared in 75 mM Na-HEPES pH 7.5, 1.5 mM EGTA, 1.5 mM MgCl$_2$, KCl 150 mM, 15% glycerol, 0.1% NP-40, protease inhibitor cocktail (Calbiochem, 539134), with 30 min 11,000 × g centrifugation. Three hundred micrograms of cell extract were incubated with 10 μl slurry α-GFP antibody conjugated to agarose beads (MBL) for 2 h, at 4 °C, with gentle agitation on wheel. After supernatant removal, beads were washed 4 times with 1 ml lysis buffer, and Laemmli sample buffer was added to the beads for SDS–PAGE and immunoblotting analysis.

For experiments of Supplementary Fig. 4g full-length human GFP-NuMA and NuMA-3xFLAG were cloned into a pCDH vector. HEK293T cells cultured as above, were co-transfected with 10 μg pCDH-GFP-NuMA plasmidic DNA and 1 μg pCDH-NuMA-3xFLAG plasmidic DNA. After 48 h from transfection, cells were treated for 16 h with 0.33 μM nocodazole. Mitotic lysates were prepared as above. Five hundred micrograms of cell extract were incubated with 10 μl slurry α-GFP antibody conjugated to agarose beads, washed 4 times in lysis buffer, and resuspended in Laemmli sample buffer for SDS-PAGE separation and immuno-blotting.

**Immunofluorescence**. For immunofluorescence, HeLa cells were plated on 13 mm coverslips coated with 5 μg/ml fibronectin. To visualize NuMA and p150$^{Glued}$ at the cortex, cells were fixed with methanol at −20 °C for 10 min. To detect LGN, α-tubulin and γ-tubulin, cells were fixed with 4% paraformaldehyde for 10 min at room temperature, followed by permeabilization with 0.3% Triton X-100 in PBS for 5 min. For all conditions, blocking was performed with 3% BSA in PBS for 1 h at room temperature. Cells were stained with mouse anti-LGN (1:5, monoclonal, Mapelli lab), mouse anti-NuMA (1:3000, monoclonal, Mapelli lab), mouse anti-p150$^{Glued}$ (1:600, BD #610473), rabbit anti-α-tubulin (1:50, Abcam #ab4074), mouse anti-α-tubulin (1:200, Sigma-Aldrich #T5168), mouse-anti-FLAG (1:200, Sigma-Aldrich #F3165) or Cy3 conjugated anti-γ-tubulin (1:200, Sigma-Aldrich #C7604) in 3% BSA + 0.05% Tween-20, followed incubation with anti-mouse or anti-rabbit AlexaFluor 647 or anti-mouse AlexaFluor 488 (1:300, Jackson ImmunoResearch #715-605-152, #715-545-150).

For Caco-2 cyst staining, cells were fixed with 4% paraformaldehyde for 30 min at room temperature, followed by permeabilization with 0.5% Triton X-100 in PBS for 30 min. Blocking was performed with a buffer containing 5% BSA, 0.2% Triton X-100 and 0.05% Tween-20 in PBS for 1.5 h at room temperature. Cysts were stained with mouse anti-α-tubulin (1:100, Sigma-Aldrich #T5168) in 0.2% Triton X-100 and 0.05% Tween-20 in PBS, followed by incubation with anti-mouse AlexaFluor488 (1:100, Jackson ImmunoResearch #715-545-150). To visualize F-actin cells were incubated with TRIC-conjugated Phalloidin (1:50, Sigma-Aldrich #P1951) for 1 h at room temperature. DNA was stained with DAPI.

**Immunoblotting**. For western blot analysis, HeLa and Caco-2 cells were synchronized with a single thymidine block as described above, and collected 8 h after the release. Cells were then lysed in lysis buffer containing 75 mM Hepes pH 7.5, 1.5 mM, EGTA, 1.5 mM MgCl2, 150 mM KCl, 0.1% NP40 and 15% Glycerol and protease inhibitors (Calbiochem, 539134). 50 μg of cell lysates were resolved by SDS-electrophoresis and transferred on a nitrocellulose membrane. Blocking was performed in TBS containing 0.1% Tween-20 and 5% low fat milk. Primary

antibody incubation was performed at room temperature for 2 h with the following dilution: anti-LGN (1:500, Mapelli lab), anti-NuMA (1:200, Mapelli lab), anti-Vinculin 1:10000 (in-house IEO), anti-α-tubulin (1:600, Abcam #ab4074), anti-GFP (1:1000, in-house IEO) and anti-FLAG (1:8000, Sigma-Aldrich #F7425).

**Microscopy.** Confocal images shown in Figs. 3, 5d, and supplementary Fig. 4, 5 were acquired on a Leica SP2 AOBS confocal microscope controlled by Leica confocal software. For HeLa cells analysis, a ×63 oil-immersion objective lens (HCX Plan-Apochromat ×63 NA 1.4 Ldb Bl) was used. For Caco-2 cyst multilumen experiments, a ×20 objective lens (HC PL FLUOTAR ×20 0.5 DRY) was used. For Caco-2 spindle angle analysis, a ×40 objective lens (HC PL Apochromat 40X NA 1.30 CS2) was used. Images shown in Fig. 5b and Supplementary Fig. 3 were acquired on a Leica SP8 confocal microscope controlled by a Leica confocal software. For cells analysis, a ×63 oil-immersion objective lens (HC PL Apochromat ×63 NA 1.4 CS2) was used. All images were processed using the software Fiji[40].

**Spindle orientation analysis.** Mitotic spindle orientation was monitored on HeLa and Caco-2 cells synchronized in metaphase. HeLa cells were plated on fibronectin-coated coverslips and stained with γ-tubulin to visualize poles and DAPI to visualize DNA. Cells were imaged in $x$–$z$ optical sections passing through the spindle poles. To determine the orientation of metaphase spindle, the angle formed by a line passing through the spindle poles and the substratum was measured exploiting the angle tool of the software Fiji. Spindle orientation analysis in Caco-2 cysts were conducted as described in Jaffe, JCB 2008[41]. Briefly, three $x$–$y$ confocal sections of the equatorial region of the cyst were acquired and then merged, in order to visualize both the spindle poles. To analyze the spindle axis orientation, the angle formed by a line passing through the spindle poles and the centroid of the cyst marked by Phalloidin was determined using the software Fiji. Statistical analysis of angle distributions was performed in Prism with the Krustal–Wallis test.

**Quantification of cortical and polar fluorescence intensity.** To quantify LGN, NuMA and p150[Glued] signal at the cell cortex, confocal sections of metaphase cells were analyzed as follow. Using the software Fiji, a 30-pixel-wide line was drawn from the spindle poles to the cell cortex, to obtain the intensity profile along the line. Using the software Matlab, the amount of protein at the cortex was calculated by integrating the profile of a 10 pixel-wide area of the peak, while the amount of protein in the cytoplasm was calculated by integrating a 10 pixel-wide area, 5-pixel distant from the peak. In Fig. 3h–l cortex/cytoplasm ratio is shown.

To quantify the fluorescence intensity of mCherry-NuMA at the spindle poles, confocal section of metaphase cell stained with α-tubulin were analyzed with the software Fiji. In details, per each cell imaged, the α-tubulin signal of one pole in focus was used to build a tubulin mask, and the mCherry signal inside the mask was integrated. A tubulin mask of the same dimension was positioned in the cytoplasm to obtain the mCherry intensity in the cytoplasm. In Supplementary Figs. 4f and 5g the pole/cytoplasm ratio of mCherry-NuMA wild-type or NuMA-ΔMT mutant is shown. For both the cortex/cytoplasm ratio and the pole/cytoplasm ratio, statistical analysis of the data was performed in Prism.

To visualize dynein recruitment at the cortex, human dynein light-intermediate chain 1 (LIC1) was cloned in a pCDH lentiviral vector in frame with a 3xFLAG tag, and transfected in HeLa cells ablated of endogenous LGN and stably expressing LGN-WT or LGN-ΔOLIGO. Transfected cells were analyzed with the software Fiji and defined as cortical LIC when a crescent of LIC was visible at the cortex. Statistical analysis of data was performed in Prism with the Fisher's exact test.

**Reporting summary.** Further information on research design is available in the Nature Research Reporting Summary linked to this article.

## Data availability
Data supporting the findings of this manuscript are available from the corresponding author upon reasonable request. A reporting summary for this Article is available as a Supplementary Information file. The source data underlying Figs. 3c-d-f-h-j-l, 5c-e and Supplementary Figs. 3e-f, 4d-f, 5e-g are provided as a Source Data file. Coordinates and structure factors have been deposited in the Protein Data Bank under the accession code PDB 6HC2.

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

## Acknowledgements

We are grateful to Alice Douangamath and Pierre Aller for their precious help in crystallization and data collection. We thank the scientists at the I04 and I04-1 beamlines of Diamond, at the X06DA beamline at the Swiss Light Source, and at the European Synchrotron Radiation Facility beamlines for valuable support in data collection. We thank M. Faretta, S. Freddi, and the IEO Imaging Unit for the use of the optical microscopes. We are grateful to Andrea Musacchio for sharing the Ndc80-Bonsai cDNA. We thank Andrea Disanza for suggestions with the cell biology, and Francesca Cella for helpful discussions. We thank Jan Faix for the generous gift of anti-GFP antibodies. We are grateful to all members of the Mapelli laboratory and to Anthony Roberts for scientific discussion, and for careful reading of the manuscript. This work was supported by grant to M.M. from the Italian Association for Cancer Research (AIRC) (IG 18629) and the Ministry of Health (RF-2013-02357254).

## Author contributions

L.P. and C.G. conducted the cell biology experiments; M.C., A.A., S.M., V.C., and F.R. performed the biochemical experiments; S.C. and J.F. have processed, combined, and merged individual diffraction datasets to produce the final complete dataset; S.C. has crystallized the protein and solved the structure; S.P. refined and illustrated the structure; M.M. supervised the project and wrote the manuscript.

## Additional information

**Competing interests:** The authors declare no competing interests.

