## [Peer Review File · Nature Communications]

Reviewers' Comments:

Reviewer #1:

Remarks to the Author:

The manuscript by Pirovano et al provides important new insights into how NuMA and LGN contribute to positioning the spindle during asymmetric division of cells. The authors report a crystal structure of NuMA and LGN that reveals an unexpected multimerization. Using a longer NuMA fragment than previous studies the authors showed that 3 NuMA molecules and 3 LGN TPR domains come together to form a heterohexamer. The authors provide a nice description of how the heterohexamer forms and use the structure to design constructs that don't oligomerize. They make use of a number of elegant cell assays to show that non-oligomerizing mutants can still recruit dynactin (and presumably dynein) to the cortex, but that asymmetric division no longer occurs properly. The manuscript also shows that a microtubule binding region of NuMA is essential for the proteins function.

The manuscript is highly suitable for Nature Communications and I strongly support its publication. I have two major comments, and a number of minor comments that may be of use to improve the manuscript and can mostly be addressed by textual changes.

Major Comment

1) The authors have provided strong evidence that the heterohexameric structure they observe is relevant for NuMA function. However I think the evidence for further oligomerization is not quite as strong. The biochemical assays clearly show that higher order oligomers exist over gel filtration, but the cellular experiments (using the Aurora-A inhibitor MLN8237) ask what happens when the heterohexamer is disrupted. I.e. They don't directly test the importance of higher order formation or whether this exists in the cell. I think this can be addressed by toning down the text.

2) The authors show that the non-oligomerizing mutant still localizes p150 to normal levels in Hela cells. I think it would be helpful to also show that dynein recruitment levels are normal. In the discussion the authors refer to dynein/dynactin recruitment, but at the moment they have only show dynactin recruitment.

Minor comments

The manuscript would generally benefit from some editing to check the grammar.

1) Line 75: "These observations suggest that NuMA is not transported to the cortex along astral MTs by dynein/dynactin". I wouldn't expect NuMA to be transported to cortex by dynein/dynactin as this is the kinesin transport direction.

2) Line 94: "lines within the inner side of the helical N-terminal TPR scaffold of LGN" – remove "within"

3) Line 86: "However, PIP2-binding promotes an additional cortical NuMA accumulation occurring in anaphase, upon Cdk1 inactivation, that supports spindle elongation and sister chromatid separation." Please add a reference.

4) Line 112: "To start investigating the architecture of force generators, we reconstituted the NuMA:LG N interaction from bacterial sources using human proteins, starting from the previously identified binding interfaces" – This sentence is difficult to understand. I think you mean "reconstituted using human proteins expressed in bacteria"

5) Line 115: "LGN1-409 (LG NTPR in the following), bound to the NuMAPEPT was consistent with a 1:1 binary interaction, in agreement with the structural data" - what structural data are the

authors referring to here? Please explain more.

My impression is that it would be more helpful to spell out again what residues are involved in the NuMA construct (e.g. NuMA1900-1928) rather than just using NuMApept.

6) Line 117: "NuMA fragment encompassing residues 1821-2001 assembles with LGNTPR a complex eluting as a single peak much earlier than the predicted molecular weight"
Doesn't make sense to me. How about something like "assembles with LGNTPR to form a complex that elutes as a single peak. This complex elutes much earlier than its expected molecular weight."

7) Line 131: As a suggestion try something like " Datasets were collected from multiple crystals and merged to produce a combined dataset with X% completeness and a I/sigma (CC0.5) of ... (See Methods for details)"

8) Line 140: I suggest the edit in capitals "In such an arrangement, the flexible NuMA chains thread in between two adjacent LGN subunits, and then line in the internal groove of the TPR domain, IN A SIMILAR WAY TO THAT observed in the NuMAPEPT:LGNTPR binary complex"

9) Line 142-146: Difficult to understand – needs rewriting

10) Line 148: consider "containing the" rather than "contributed by the"

11) Line 160: consider "mutated" rather than "substituted"

12) Line 163: consider "prevents hexamer" (rather than prevent hexamers)

13) Line 165: State which residues are missing.

14) Line 166: I would have found it more useful to say the residues where the density becomes visible (rather than using NuMApept).

15) Lines 167-171. I don't understand why a construct lacking 1881-1897 running at the same size as the NuMAIgnbd supports the authors model. Please could the authors explain this point in more detail?

16) Line 177: "All interfaces within the hetero-hexamer are repeated essentially identical" - remove "repeated"

17) Line 180-182: Could the authors clarify what they mean by molecular determinants? I don't understand this paragraph.

18) Line 219: Consider "cortical" rather than "cortically"

19) Line 240-242: consider saying the "LGN-binding domain overlaps with a site (give residues...) involved spindle pole activities." It seemed a bit odd to talk about exon-24.

20) Line 258: The authors need to explain when they chose to look at oligomerization of 1592-1694 (as opposed to e.g. the whole coiled coil region).

21) Line 279-280: It is not clear what this means – the authors may need to spell out what the Mw of the LGNoligo/NuMA complex would be if it didn't oligomerise.

22) Line 280: consider "confirms" not "confirm"

23) Line 303: The authors should give the residue numbers for the MT binding domain.

24) Why did the authors choose to use the whole section from 1821-2115, when the MT binding region is C-terminal to the LGN binding region?

25) Line 394: consider "result" rather than "results".

Reviewer #2:

Remarks to the Author:

Proper orientation of the mitotic spindle is fundamental for appropriate cell division, and this requires the interaction between the astral microtubules and the cell cortex. It is well established

that the evolutionarily conserved tripartite complex in human cells (Gai1-3/LGN/NuMA) cortically anchors minus-end motor dynein complex at the cell cortex. It is the dynein activity together with the impact of NuMA on the dynamic astral microtubules that is required to maintain proper spindle orientation during metaphase. However, the stoichiometry and the structure of this complex involved in the generation of the pulling forces is far from complete.

In this manuscript, the authors attempted to study the crystal structure of N-ter TPR motifs containing a longer fragment of LGN bound to the shorter C-ter fragment of NuMA and uncovered that such combinations of LGN and NuMA fragments form hetero-hexamers with a 3:3 stoichiometry. Based on the initial structure the authors went ahead and analyzed the importance of the hetero-hexamers for spindle orientation *in vivo* by utilizing Caco-2 cells as well as HeLa. Importantly, the authors uncovered that expression of LGN lacking the oligomerization domain recruit equal amount of cortical NuMA, however, perturbs proper spindle orientation in cells lacking endogenous LGN. Conversely, they discovered that NuMA Δ OLIGO expression also impacts spindle orientation in cells lacking endogenous NuMA. Additionally, the authors reveal that the dimerization coiled-coil region of NuMA is critical for making higher oligomers and proposed that in cells that dimeric NuMA is associated with multimeric NuMA-LGN hetero-hexamers for proper spindle orientation.

This manuscript by Pirovano et al., is interesting as the authors have attempted to characterize the role of multi-order assemblies of the components of the tripartite complex in localizing dynein and generating pulling forces, which is not so well characterized at the structure-functional level. However, based on their experimental data I am not convinced by the conclusions drawn by the authors (see major concerns for details). Also, it is not clear to me whether the full-length proteins when expressed in their native levels forms such higher-order oligomers at the cell cortex to generate sufficient pulling forces for proper spindle orientation. Therefore, I feel that the current draft of the manuscript needs the substantial amount of work before it is suitable for publication.

Major Concerns:

1. Pirovano et al., have analyzed oligomer assembly of LGN(7-367) and NuMA (1864-1928) by crystallography and they found that such fragments are assembled in a hetero-hexamers configuration (Figure 2). However, there is no evidence whatsoever provided by the authors that full-length proteins in their native configuration also forms higher-oligomers. Recently Okumura et al., 2018, have shown that C-ter of a mutated NuMA fragment (mutated at the Cdk1-phosphorylation residues) could form NuMA clusters at the cell membrane. However, whether the full-length NuMA could form such oligomers especially at the cell cortex is not clear, and thus it would be crucial to show that NuMA, when expressed in their native levels, can form higher-order oligomer at the cell cortex and that is gone in mutations affecting hetero-hexamers formation.

2. The data from the Caco-2 cells is interesting and the authors show that LGN- Δ OLIGO is unable to rescue the multi-lumen phenotype when expressed upon depletion of the endogenous protein. Multi-lumen formation is a complex phenotype, and thus the authors should analyze the spindle orientation during metaphase upon expression of LGN as well as LGN- Δ OLIGO to see if the apparent phenotype is originated only because of defective spindle orientation. Authors must also include LGN(L54A and Y58A) mutant in their spindle orientation/multi-lumen phenotype to strengthen indeed it is the oligomerization function of LGN-NuMA interaction rather than any other uncharacterized role of LGN (1-12 and 350-366 aa residues) in spindle orientation. Therefore, I feel it is too premature to conclude that 'NuMA:LGN oligomerization is essential for planer cell division and correct cystogenesis.' Also, I wonder why the authors did not show mCherry-signal for panels 3b and 3d. It will be critical to know if the expression of LGN-mCherry or LGN- Δ OLIGO-mCherry is similar when spindle orientation is analyzed.

Ectopic expression of LGN cause spindle rotation in MDCK and HeLa. Therefore, it would be worthwhile to additionally test if ectopic expression of LGN- Δ OLIGO-mCherry fail in causing spindle

rotation in contrast to the wild-type construct.

3. As mentioned for point 2, for spindle orientation experiments shown on Supplemental Figure 3e, it would be essential to show the mCherry-NuMA levels when expressed in cells lacking endogenous NuMA.

4. It appears from Figure 3J that the overall levels of p150Glued levels are significantly down both at the cortex as well as in the cytoplasm and that could be the reason for the unchanged ratio of cortical/cytoplasm in expressing LGN- Δ OLIGO. I would also encourage the authors to analyze GFP-DHC-1 signal in such a setting.

5. In Figure 4, by analyzing various fragments of NuMA and LGN using biochemical means, the authors have shown the coiled-coil of NuMA could be the key for the higher oligomer formation. Therefore, I wonder what is the impact of NuMA Δ 1592-1694 on cortical LGN localization and spindle orientation.

6. In Figure 5, the authors made an interesting observation that expression of NuMA-GoLoco can rescue the spindle orientation phenotype achieved by Aurora A inactivation, however, NuMA- Δ OLIGO-GoLoco that localize at the cell cortex similar to NuMA-GoLoco is unable to rescue spindle orientation. NuMA- Δ OLIGO-GoLoco lacks ~ 40 aa, and thus it may well be that non-rescue of the spindle orientation phenotype stems from the inability of the NuMA- Δ OLIGO-GoLoco to perform (uncharacterized) functions other than its impact on the in vitro oligomerization assemblies. Therefore, I would also analyze the impact of NuMA- Δ OLIGO-GoLoco expression on the astral microtubules and dynein localization in such experimental settings.

Minor points:

1. In general, the current draft does provide the detailed information regarding the exact nature of the constructs in the Figure legends (whether GFP or mCherry-tagged). The readers need to move backward and forward from methods/main text to the Figure legends to check which plasmid/construct was utilized. For instance, in the Figure legend for Figure 5, the authors forgot to provide information related to the NuMA- Δ OLIGO-GoLoco that is the central construct for this piece of data.

2. The title of the authors is quite misleading since to my view it is not established whether in the cells under native conditions endogenous NuMA and LGN make Hexameric structures that are required for dynein-dependent spindle orientation. Therefore, as explained earlier, I would suggest the authors to evaluate this more critically by performing experiments (biochemically or cell biology-based) to test this using cellular assays.

3. In experiments performed with Caco-2 cells, I would suggest using an alternate apical marker (e.g. ZO-1).

4. In the Western blot on Supplementary Figure 3a and 3, it is not clear which band represent LGN since other bands are also affected by RNAi-mediated depletion.

5. In S3g the cell expressing NuMA Δ LGNBD has tilted spindle, and that is why the spindle-poles are not visible that could affect spindle pole quantification. I would suggest quantifying spindle-poles in cells where the spindle-poles are clearly visible.

Reviewer #3:

Remarks to the Author:

This paper addresses how the NuMA regulator LGN assembles complexes on the cortex that are

structurally functional for spindle orientation. The foundation of the paper is the finding that fragments of NuMA larger than those previously studied interact with LGN form a 3:3 complex between the proteins, as opposed to the 1:1 complex of the smaller fragment. A low resolution structure of the larger complex, determined by x-ray crystallography, is included in the paper. The structure, along with deletion analysis, allowed the authors to create mutations that selective disrupt the oligomerization function and they find that LGN proteins harboring these mutations are unable to rescue the spindle orientation defect found in cells that lack LGN. Overall this is an important contribution because it identifies a new LGN-NuMA contact point and furthers our understanding of the stoichiometry of the interaction and its functional importance. While it does leave the question of why oligomerization is required unanswered, hopefully this will be a topic for further investigation.

Minor comments:

Please label the dot plots and bar graphs through the paper (e.g. Figure 3c, e, ...) so that the relevant condition is underneath each plot

The following statement is incorrect unless the attached reference demonstrated that targeting full length NuMA to the cortex is sufficient for spindle orientation: "light-induced ectopic delivery of this NuMA fragment to the cortex results in dynein/dynactin recruitment but cannot support spindle pulling⁹, implying that during spindle orientation additional functions encoded by the C-terminal cargo binding portion of NuMA are essential for the spindle orientation process."

Reviewers' comments:

Reviewer #1 (Remarks to the Author):

The manuscript by Pirovano et al provides important new insights into how NuMA and LGN contribute to positioning the spindle during asymmetric division of cells. The authors report a crystal structure of NuMA and LGN that reveals an unexpected multimerization. Using a longer NuMA fragment than previous studies the authors showed that 3 NuMA molecules and 3 LGN TPR domains come together to form a heterohexamer. The authors provide a nice description of

how the heterohexamer forms and use the structure to design constructs that don't oligomerize. They make use of a number of elegant cell assays to show that non-oligomerizing mutants can still recruit dynactin (and presumably dynein) to the cortex, but that asymmetric division no longer occurs properly. The manuscript also shows that a microtubule binding region of NuMA is essential for the proteins function.

The manuscript is highly suitable for Nature Communications and I strongly support its publication. I have two major comments, and a number of minor comments that may be of use to improve the manuscript and can mostly be addressed by textual changes.

Major Comment

1) The authors have provided strong evidence that the heterohexameric structure they observe is relevant for NuMA function. However I think the evidence for further oligomerization is not quite as strong. The biochemical assays clearly show that higher order oligomers exist over gel filtration, but the cellular experiments (using the Aurora-A inhibitor MLN8237) ask what happens when the heterohexamer is disrupted. I.e. They don't directly test the importance of higher order formation or whether this exists in the cell. I think this can be addressed by toning down the text.

We agree with the Reviewer that our cellular assays support the notion that NuMA:LGN hexamers are essential for spindle orientation in cultured HeLa cells and polarized Caco-2 cysts, but do not provide direct evidence for the existence of cortical networks of dynein/dynactin in cells. The idea that NuMA:LGN oligomers could organize dynein motors at the cortex stems from our *in vitro* findings that dimeric NuMA constructs encompassing the LGN-binding domain are able to organize large protein assemblies compatible with the presence of NuMA:LGN hexamers joint by NuMA coiled-coils. We have toned down the text to underscore that the hypothesis of dynein cortical networks still needs to be demonstrated in mitotic cells.

2) The authors show that the non-oligomerizing mutant still localizes p150 to normal levels in HeLa cells. I think it would be helpful to also show that dynein recruitment levels are normal. In the discussion the authors refer to dynein/dynactin recruitment, but at the moment they have only show dynactin recruitment.

We agree with the Referee that we do not have experimental data showing the distribution of dynein in cells expressing the oligomerization-deficient LGN- Δ OLIGO rescue construct. Following the Reviewer suggestion, we have tried to visualize cortical dynein by immuno-fluorescence using commercially available antibodies against dynein intermediate chain (Merck-Millipore, catalogue number MAB1618). However, we could only detect weak cortical signals that could not be quantified reliably (see enclosed Figure 1). Therefore, we decided to transfect HeLa cells with a FLAG-tagged version of dynein light intermediate chain (human LIC1) and use it as a marker of dynein localization in mitotic cells lacking endogenous LGN and expressing the LGN rescue constructs. Representative images of this experiment and relative quantifications have been added in the Supplementary Figure 3d-e. They show that in metaphase, cortical dynein is lost upon LGN

ablation, and its levels are fully restored in cells expressing LGN-wild-type or LGN- Δ OLIGO, consistently with the localization of dynein and NuMA.

Figure 1. Immuno-staining of HeLa cells with anti-dynein intermediate chain antibodies (Merck-Millipore, cat. MAB1618). Representative confocal sections of mitotic HeLa cells wild-type or lacking LGN and expressing the rescue constructs LGN-WT-mCherry or LGN- Δ OLIGO-mCherry. DNA is visualized by DAPI (cyan), dynein intermediate chain (dynein DIC) is shown in white.

Minor comments

The manuscript would generally benefit from some editing to check the grammar.

1) Line 75: “These observations suggest that NuMA is not transported to the cortex along astral MTs by dynein/dynactin”. I wouldn’t expect NuMA to be transported to cortex by dynein/dynactin as this is the kinesin transport direction.

The Reviewer is right. We have corrected the text replacing “dynein” with “kinesins”.

2) Line 94: “lines within the inner side of the helical N-terminal TPR scaffold of LGN” – remove “within”

We have removed “within”.

3) Line 86: “However, PIP2-binding promotes an additional cortical NuMA accumulation occurring in anaphase, upon Cdk1 inactivation, that supports spindle elongation and sister chromatid separation.” Please add a reference.

We agree with the Reviewer that references for the direct NuMA binding to the plasma membrane in anaphase were missing. We have added them.

4) Line 112: “To start investigating the architecture of force generators, we reconstituted the NuMA:LGN interaction from bacterial sources using human proteins, starting from the previously

identified binding interfaces” – This sentence is difficult to understand. I think you mean “reconstituted using human proteins expressed in bacteria”

We have rephrased the sentence to as suggested by the Reviewer to make it clearer.

5) Line 115: “LGN1-409 (LGNTPR in the following), bound to the NuMAPEPT was consistent with a 1:1 binary interaction, in agreement with the structural data” - what structural data are the authors referring to here? Please explain more.

Here we refer to the crystallographic structure of the TPR domain of mouse LGN (residues 15-350) in complex with a short peptide of NuMA spanning residues 1899-1926, which corresponds to the NuMA fragment we name NuMA^{PEPT} (1900-1928) in the original manuscript. The structure was described by Zhu and collaborators in a Mol. Cell article in 2011, reference 19 in our bibliography. To clarify the statement, we have rewritten the paragraph.

My impression is that it would be more helpful to spell out again what residues are involved in the NuMA construct (e.g. NuMA1900-1928) rather than just using NuMApept.

We thank the Reviewer for the comment. In the initial submission, we decided to introduce the term NuMA^{PEPT} with the idea that indicating the residue number explicitly would have made the text less fluent. We have now replaced in the manuscript “PEPT” with “1900-1928” as suggested.

6) Line 117: “NuMA fragment encompassing residues 1821-2001 assembles with LGNTPR a complex eluting as a single peak much earlier than the predicted molecular weight” Doesn’t make sense to me. How about something like “assembles with LGNTPR to form a complex that elutes as a single peak. This complex elutes much earlier than its expected molecular weight.”

We thank the Referee for the suggestion, we have modified the text as indicated.

7) Line 131: As a suggestion try something like “ Datasets were collected from multiple crystals and merged to produce a combined dataset with X% completeness and a I/sigma (CC0.5) of ... (See Methods for details)”

We thank the Referee for the suggestion, we have modified the text as indicated.

8) Line 140: I suggest the edit in capitals “In such an arrangement, the flexible NuMA chains thread in between two adjacent LGN subunits, and then line in the internal groove of the TPR domain, IN A SIMILAR WAY TO THAT observed in the NuMAPEPT:LGNTPR binary complex”

We thank the Referee for the suggestion, we have modified the text as indicated.

9) Line 142-146: Difficult to understand – needs rewriting

We have rewritten the paragraph trying to make clearer the evidence that the 8 TPRs of LGN arrange in a bent super-helix due to the presence of the non-canonical TPR-4, whose antiparallel helices are 10-residue longer than the helices of canonical TPRs (we have already described this peculiar feature of LGN in a previous manuscript by Culurgioni et al., PNAS 2011). In the LGN:NuMA hexamers, the curvature of the TPR array of LGN is more pronounced due to the contacts between the N-terminal helix of one LGN-TPR molecule and TPR-8 of the adjacent LGN-TPR in the donut, as shown in Supplementary Fig. 2.

10) Line 148: consider “containing the” rather than “contributed by the”

We have modified the text as indicated.

11) Line 160: consider “mutated” rather than “substituted”

We have modified the text as indicated.

12) Line 163: consider “prevents hexamer” (rather than prevent hexamers)

We have modified the text as indicated.

13) Line 165: State which residues are missing.

We have modified the text as indicated.

14) Line 166: I would have found it more useful to say the residues where the density becomes visible (rather than using NuMApept).

We have modified the text as indicated.

15) Lines 167-171. I don't understand why a construct lacking 1881-1897 running at the same size as the NuMA^{lgnbd} supports the authors model. Please could the authors explain this point in more detail?

We thank the Reviewer for the comment. The modest resolution of the data (4.3 Å) made it difficult to trace the NuMA residues not already present in the model used for molecular replacement, i.e. residues 1861-1899 before the NuMA-PEPT (1900-1928). Based on the good quality of the electron density map (see Figure 2f), we built the NuMA stretch 1861-1880 and left unmodeled residues 1881-1897, for which we could not see additional electron density, likely because these residues are in a flexible loop. Importantly, the NuMA residues that we modelled in the electron density bridge between two adjacent TPR molecules, this way stabilizing the hexameric assembly (Figure 2d and 2f). To confirm the map interpretation, we decided to generate a NuMA deletion mutant lacking the unmodeled residues, that we named NuMA^{LGNBD-Δ1881-1897}. We reasoned that if our map interpretation was correct, the oligomeric state of LGN-TPR:NuMA^{LGNBD-Δ1881-1897} complexes would have been unchanged (and the complex would have eluted from SEC at the same

volume of LGN-TPR:NuMA^{LGNBD} complexes). Conversely, if the visible electron density corresponded to the NuMA residues 1881-1897 and the N-terminus of the construct 1861-1880 was the invisible region, we would have observed a 1:1 LGN-TPR:NuMA^{LGNBD-Δ1881-1897} complex eluting from SEC as the LGN-TPR:NuMA¹⁸⁷⁷⁻¹⁹²⁸ complex. The experiment presented in Supplementary Fig. 2b showed that LGN-TPR:NuMA^{LGNBD-Δ1881-1897} complexes are hexamers, demonstrating that our map interpretation is correct. We understand that this reasoning was somehow implicit in the text, and we have rephrased the paragraph to make it more explicit.

16) Line 177: “All interfaces within the hetero-hexamers are repeated essentially identical” - remove “repeated”

We have removed “repeated”.

17) Line 180-182: Could the authors clarify what they mean by molecular determinants? I don't understand this paragraph.

With the term “molecular determinants” we wanted to indicate side chains and residue stretches on NuMA and LGN that are essential for the assembly of LGN:NuMA hetero-hexamers and whose mutation results in 1:1 complexes. We have rewritten the sentence to make it clearer.

18) Line 219: Consider “cortical” rather than “cortically”

We have modified the text as suggested.

19) Line 240-242: consider saying the “LGN-binding domain overlaps with a site (give residues...) involved spindle pole activities.” It seemed a bit odd to talk about exon-24.

We understand the point, and have modified the text as suggested.

20) Line 258: The authors need to explain when they chose to look at oligomerization of 1592-1694 (as opposed to e.g. the whole coiled coil region).

We thank the Reviewer for the observation. To assess the oligomeric state of NuMA coiled-coil regions longer than the NuMA₁₅₉₂₋₁₆₉₄ shown in Figure 4c, we designed longer NuMA constructs based on coiled-coil prediction, spanning residues 216-1692, 416-1692 and 706-1692. Unfortunately, these constructs showed a tendency to degrade and aggregate, as shown in the SEC profiles of the enclosed Figure 2. As we could not purify homogeneous and monodisperse samples of these constructs for Static-Light-Scattering analysis, we decided to test the oligomeric state of NuMA full-length in cells by co-immunoprecipitation (co-IP) experiments. To this aim, we co-transfected HEK293 cells with plasmid coding for FLAG-tagged and GFP-tagged versions NuMA, and tested the presence of FLAG-NuMA immunoprecipitating with GFP-NuMA in mitotic lysates. As shown in Suppl. Fig. 4g, FLAG-NuMA co-IP with GFP-NuMA, indicating that dimers of differentially tagged full-length proteins exists in cells.

Figure 2. Biochemical analysis of NuMA coiled-coil constructs. SEC elution profiles (left) and Coomassie-stained SDS-PAGE of the corresponding peak fractions (right) of three constructs of NuMA encompassing different portions of the coiled-coil region. The constructs were designed on the basis of *in silico* predictions, cloned as His-tagged proteins and purified by affinity and ion-exchange chromatography prior to SEC analysis.

21) Line 279-280: It is not clear what this means – the authors may need to spell out what the Mw of the LGNoligo/NuMA complex would be if it didn't oligomerise.

We have rephrased the sentence to clarify that LGN-TPR- Δ OLIGO forms a 1:1 complex with one NuMA chain. However, as the NuMA-chimera construct dimerizes via the coiled-coil region 1592-1694 (Figure 4c), two molecules of LGN-TPR- Δ OLIGO can bind to NuMA-chimera, regardless of the NuMA:LGN oligomerization, resulting in a 2:2 hetero-tetramer with a theoretical molecular mass of about 142 KDa. This is in line with the 136 KDa molecular weight of the complex measured by SLS analysis (shown in Figure 4g).

22) Line 280: consider “confirms” not “confirm”

We have corrected the typo.

23) Line 303: The authors should give the residue numbers for the MT binding domain.

We have added the residue numbers of the C-terminal MT-binding domain of NuMA previously identified (Gallini et al, 2016 Curr. Biol.)

24) Why did the authors choose to use the whole section from 1821-2115, when the MT binding region is C-terminal to the LGN binding region?

We thank the Referee for the observation. We have repeated the MT-cosedimentation experiment with a NuMA fragment spanning residues 2002-2115, which encompasses the MT-binding domain but exclude the LGN-BD. The experiment confirms that this shorter MT-binding region of NuMA binds the MT-lattice regardless of the tubulin tails, as shown in the new Supplementary Figure 5a.

25) Line 394: consider “result” rather than “results”.

We have corrected the typo.

Reviewer #2 (Remarks to the Author):

Proper orientation of the mitotic spindle is fundamental for appropriate cell division, and this requires the interaction between the astral microtubules and the cell cortex. It is well established that the evolutionarily conserved tripartite complex in human cells (Gai1-3/LGN/NuMA) cortically anchors minus-end motor dynein complex at the cell cortex. It is the dynein activity together with the impact of NuMA on the dynamic astral microtubules that is required to maintain proper spindle orientation during metaphase. However, the stoichiometry and the structure of this complex involved in the generation of the pulling forces is far from complete.

In this manuscript, the authors attempted to study the crystal structure of N-ter TPR motifs containing a longer fragment of LGN bound to the shorter C-ter fragment of NuMA and uncovered that such combinations of LGN and NuMA fragments form hetero-hexamers with a 3:3 stoichiometry. Based on the initial structure the authors went ahead and analyzed the importance of the hetero-hexamers for spindle orientation *in vivo* by utilizing Caco-2 cells as well as HeLa. Importantly, the authors uncovered that expression of LGN lacking the oligomerization domain recruit equal amount of cortical NuMA, however, perturbs proper spindle orientation in cells lacking endogenous LGN. Conversely, they discovered that NuMA Δ OLIGO expression also impacts spindle orientation in cells lacking endogenous NuMA. Additionally, the authors reveal that the dimerization coiled-coil region of NuMA is critical for making higher oligomers and proposed that in cells that dimeric NuMA is associated with multimeric NuMA-LGN hetero-hexamers for proper spindle orientation.

This manuscript by Pirovano et al., is interesting as the authors have attempted to characterize the role of multi-order assemblies of the components of the tripartite complex in localizing dynein and generating pulling forces, which is not so well characterized at the structure-functional level. However, based on their experimental data I am not convinced by the conclusions drawn by the authors (see major concerns for details). Also, it is not clear to me whether the full-length proteins when expressed in their native levels forms such higher-order oligomers at the cell cortex to generate sufficient pulling forces for proper spindle orientation. Therefore, I feel that the current

draft of the manuscript needs the substantial amount of work before it is suitable for publication.

Major Concerns:

1. Pirovano et al., have analyzed oligomer assembly of LGN(7-367) and NuMA (1864-1928) by crystallography and they found that such fragments are assembled in a hetero-hexamers configuration (Figure 2). However, there is no evidence whatsoever provided by the authors that full-length proteins in their native configuration also forms higher-oligomers. Recently Okumura et al., 2018, have shown that C-ter of a mutated NuMA fragment (mutated at the Cdk1-phosphorylation residues) could form NuMA clusters at the cell membrane. However, whether the full-length NuMA could form such oligomers especially at the cell cortex is not clear, and thus it would be crucial to show that NuMA, when expressed in their native levels, can form higher-order oligomer at the cell cortex and that is gone in mutations affecting hetero-hexamers formation.

We thank the Reviewer for the observation. In general, biochemical and structural characterization of multi-subunit complexes performed *in vitro* provide molecular details of the oligomeric state and topology of the reconstituted interactions to a resolution scale higher than that achievable in cells. Specifically, in the case of the NuMA/LGN interaction, it is not trivial to address the oligomeric state of the complexes that they form in cells because 1) the interaction occurs transiently at the cortex in mitosis, 2) the dimension of the hetero-hexamers is too small to allow visualization of individual copies of the molecules by high-resolution microscopy (such as STORM microscopy), and 3) full-length NuMA per se is dimeric and therefore also the oligomerization-deficient NuMA and LGN constructs that we designed based on the hetero-hexamer structure (i.e. LGN- Δ OLIGO and NuMA- Δ OLIGO), in the context of full-length NuMA would form 2:2 complexes. In line with these considerations, we have been trying to develop an experimental strategy that could prove the formation of higher order NuMA:LGN oligomers in cells. To bypass the dimerization of full-length NuMA, we generated a HEK293T cell line stably depleted of NuMA by sh-RNA, and expressing a monomeric C-terminal portion of NuMA encompassing the LGN-binding domain but not the coiled-coil (i.e. NuMA-1821-2115, a construct that was already characterized in a previous manuscript by Gallini et al., Curr. Biol. 2016). We then co-transfected these cells with either GFP-LGN-WT and FLAG-LGN-WT or GFP-LGN- Δ OLIGO and FLAG-LGN- Δ OLIGO, and tested whether in mitotic lysates the GFP-tagged version of LGN could immuno-precipitate the FLAG-tagged version of LGN together with NuMA-1821-2115. The experiment revealed that only GFP and FLAG-tagged LGN wild-type can form complexes with NuMA-1821-2115, while LGN- Δ OLIGO cannot (see new figure 4a). This evidence supports the notion that LGN and NuMA assemble higher-order oligomers in mitotic cells, and that the same mutations impairing oligomerization *in vitro* disrupt oligomer formation in cells, suggesting that the topology of the oligomer is the same. We understand that in our experimental setting the information regarding the localization of the LGN:NuMA complexes is lost, but we believe that the establishment of a more sophisticated experimental setting to address the stoichiometry of endogenous LGN/NuMA complexes at the cortex will require a dedicated effort in the future.

2. The data from the Caco-2 cells is interesting and the authors show that LGN- Δ OLIGO is unable to rescue the multi-lumen phenotype when expressed upon depletion of the endogenous protein.

Multi-lumen formation is a complex phenotype, and thus the authors should analyze the spindle orientation during metaphase upon expression of LGN as well as LGN- Δ OLIGO to see if the apparent phenotype is originated only because of defective spindle orientation. Authors must also include LGN(L54A and Y58A) mutant in their spindle orientation/multi-lumen phenotype to strengthen indeed it is the oligomerization function of LGN-NuMA interaction rather than any other uncharacterized role of LGN (1-12 and 350-366 aa residues) in spindle orientation. Therefore, I feel it is too premature to conclude that ‘NuMA:LGN oligomerization is essential for planer cell division and correct cystogenesis.’ Also, I wonder why the authors did not show mCherry-signal for panels 3b and 3d. It will be critical to know if the expression of LGN-mCherry or LGN- Δ OLIGO-mCherry is similar when spindle orientation is analyzed. Ectopic expression of LGN cause spindle rotation in MDCK and HeLa. Therefore, it would be worthwhile to additionally test if ectopic expression of LGN- Δ OLIGO-mCherry fail in causing spindle rotation in contrast to the wild-type construct.

We thank the Reviewer for the comments. As suggested, we have analyzed the spindle orientation during cystogenesis of Caco-2 cells lacking endogenous LGN and expressing LGN full-length or the LGN- Δ OLIGO. The analysis showed that upon LGN loss the metaphase spindle axis is misoriented compared to normal planar divisions. Correct spindle orientation is rescued by expression of LGN wild-type but not of the LGN- Δ OLIGO mutant, confirming that the multi-lumen phenotype observed in Caco-2 cysts expressing the oligomerization-deficient LGN construct can be ascribed to spindle orientation defects. These results are shown in Figure 3d and Supplementary Figure 3c.

We take the Referee’s point that in principle using an LGN point mutant abrogating NuMA-LGN oligomerization for rescue experiments would be preferential than using a mutant lacking the entire oligomerizing stretches of LGN encoded by residues 1-12 and 350-366. However, the biochemical analyses shown in Supplementary Figure 2a revealed that LGN-L54A-Y58A only partly recapitulate the oligomerization deficient phenotype, because LGN-TPR-L54A-Y58A is still able to assemble hetero-hexamers eluting from a SEC column at high molecular weight, although with a reduced efficiency. For this reason, and because to our knowledge no known functions of LGN are mediated by the oligomerizing stretches that we remove, to assess the relevance of NuMA-LGN oligomerization in spindle orientation we have preferred to use a the fully penetrant mutant LGN- Δ OLIGO.

The Caco-2 and HeLa cells used for spindle cystogenesis and spindle orientation analysis of Figure 3b and 3e stably express LGN-mCherry constructs are under puromycin selection, at levels comparable with the endogenous protein as shown in the blots of Supplementary figure 3a and 3b. At single cell level, the quantifications of cortical LGN levels in figure 3g-h show that the mCherry-LGN rescue constructs are expressed in HeLa cells at the same level of endogenous LGN. In this experiment, we have decided to quantify the cortical LGN signal using anti-LGN antibodies to compare the amount of the LGN-mCherry rescue constructs to that of the endogenous protein, and to show that the sh-RNA targeting LGN abrogate cortical crescents of endogenous LGN. For the sake of completeness, we present in the enclosed Figure 3 representative images of mitotic HeLa and Caco-2 cells expressing LGN-mCherry and LGN- Δ OLIGO-mCherry showing that the cortical levels of the mCherry signal are comparable in the four cell lines.

a**b**
Figure 3. Cortical levels of mCherry signal in HeLa and Caco-2 cell lines expressing LGN-mCherry constructs. Confocal sections of mitotic HeLa (a) and Caco-2 (b) cell lines lacking endogenous LGN and expressing the rescue constructs LGN-WT-mCherry and LGN- Δ OLIGO-mCherry. For this experiment, Caco-2 cells were grown in monolayer. DNA is visualized by DAPI (cyan), the mCherry signal is shown white.

We thank the Reviewer for the experimental suggestion regarding spindle rotation. To test whether over-expression of LGN- Δ OLIGO triggers excessive spindle rocking as observed for the over-expression to LGN wild-type, we imported the same protocols used by Kotak and colleagues (Kotak, JCB 2012), and analysed the oscillations of the metaphase plate in HeLa cells stably expressing H2B-GFP transfected with LGN-WT-mCherry or LGN- Δ OLIGO-mCherry (see enclosed Figure 4). This analysis showed that over-expression of LGN- Δ OLIGO induces spindle rocking to the same extent of that observed upon overexpression of LGN wild-type (Fig. 4a enclosed). We reasoned that this result is not surprising because the LGN constructs are massively overexpressed in transiently transfected HeLa cells compared to the endogenous protein (Fig. 4b enclosed), likely bypassing the regulatory mechanisms governing spindle positioning under physiological conditions.

a**b**
Figure 4. Analysis of spindle oscillation in HeLa cells over-expressing LGN-WT and LGN- Δ OLIGO. (a) Images from time lapse microscopy of metaphase HeLa cells stably expressing GFP-H2B transfected with LGN-WT-mCherry or LGN- Δ OLIGO-mCherry. The GFP-H2B signal is overlaid with DIC, the position of the metaphase plate is marked with a white line. About 10 cells per condition were filmed in 2 independent experiments, taking frames every 3 minutes. The extent of oscillation was calculated considering the frequency at which the metaphase plate rotates more than 10° between two consecutive frames, and plotted in bar graphs on the right along with the SD. A two-tailed Student's test was applied to assess the statistical relevance between transfected conditions and non-transfected control cells. (b) Immuno-blot showing the levels of ectopically expressed LGN-mCherry proteins in mitotic HeLa lysates. The band of endogenous LGN is indicated with a black arrow.

3. As mentioned for point 2, for spindle orientation experiments shown on Supplemental Figure 3e, it would be essential to show the mCherry-NuMA levels when expressed in cells lacking endogenous NuMA.

We agree with the Referee. On a population scale, the levels of NuMA-mCherry proteins expressed in HeLa cells depleted of endogenous NuMA are shown in the immuno-blot of Supplementary figure 4b. The levels of the mCherry-tagged NuMA fusions are also visualized in the panels of Supplementary figure 4e, showing representative images of the ones used for quantifications of NuMA mutants at the spindle poles. As suggested, we have now added also panels showing the mCherry-signal for the spindle orientation experiments of Supplementary figure 4c (that is the panel corresponding to supplementary figure 3e in the first submission).

4. It appears from Figure 3J that the overall levels of p150Glued levels are significantly down both at the cortex as well as in the cytoplasm and that could be the reason for the unchanged ratio of cortical/cytoplasm in expressing LGN- Δ OLIGO. I would also encourage the authors to analyze GFP-DHC-1 signal in such a setting.

We understand that the Reviewer might be concerned about the p150 levels as the cortical staining shown in Fig. 3k seem less pronounced than those of LGN and NuMA, likely because of the limited quality of the anti-p150 antibody that we used. Following the Reviewer suggestion, we set out to quantify the levels of cortical dynein in the same cellular setting. Unfortunately, we could not use GFP-DHC-1 because the HeLa cell lines expressing the sh-RNA targeting LGN were generated by infection with a lentivirus containing a GFP reporter. Thus, we decided to transfect cells with a FLAG-tagged version of dynein light intermediate chain 1 (human LIC1), and image cortical recruitment of LIC1 by IF with anti-FLAG antibodies. The results of these analyses have been included in the Supplementary figures 3d-e. They show that in metaphase cells, cortical dynein is lost upon LGN ablation, and its levels are rescued to the same extent of control cells in the presence of LGN-WT or LGN- Δ OLIGO. This evidence is consistent with the cortical localization of NuMA and p150Glued in the same cell lines, supporting the notion that NuMA/Dynein/Dynactin work in the same complex in spindle orientation.

5. In Figure 4, by analyzing various fragments of NuMA and LGN using biochemical means, the authors have shown the coiled-coil of NuMA could be the key for the higher oligomer formation.

Therefore, I wonder what is the impact of NuMA Δ 1592-1694 on cortical LGN localization and spindle orientation.

We thank the Referee for the observation. Indeed, the fact that NuMA is dimeric and forms hetero-hexamers with LGN is key for the assembly or high-order oligomeric networks at the cortex. In figure 4, we decided to measure by Static-Light-Scattering the oligomeric state of a short fragment of the coiled-coil of NuMA because longer fragments resulted unstable and could not be purified to homogeneity (see also Figure 1 in this rebuttal, and the answer to point 20 of Reviewer 1). Importantly, new experiments confirmed that full-length GFP-NuMA and FLAG-NuMA co-immunoprecipitate from mitotic lysates demonstrating that full-length NuMA self-assemble in cells (see new Supplementary figure 4g). Secondary structure predictions show that the coiled-coil region of NuMA encompasses a central portion of the protein from about residue 210 to residue 1694. This means that a truncated NuMA- Δ 1592-1694 mutant would still be dimeric, and would not be suited to investigate the relevance of NuMA dimerization in cortical targeting of LGN and spindle orientation. On the other hands, recent evidence from the Kiyomitsu lab (Okumura et al., eLife 2018) indicate that the length of the central coiled-coil region of NuMA is important for spindle positioning because a NuMA-bonsai variant lacking residues 705-1700 cannot sustain pulling forces required for spindle centering in HeLa cells in spite of localizing at the cortex (reported in figure 5 in Okumura et al., eLife 2018). For all these considerations, we reasoned that analysis of LGN localization and spindle orientation in HeLa cells expressing the rescue construct NuMA- Δ 1592-1694 would not clarify the importance of NuMA dimerization for spindle orientation. Therefore, we preferred not to perform the experiment.

6. In Figure 5, the authors made an interesting observation that expression of NuMA-GoLoco can rescue the spindle orientation phenotype achieved by Aurora A inactivation, however, NuMA- Δ OLIGO-GoLoco that localize at the cell cortex similar to NuMA-GoLoco is unable to rescue spindle orientation. NuMA- Δ OLIGO-GoLoco lacks \sim 40 aa, and thus it may well be that non-rescue of the spindle orientation phenotype stems from the inability of the NuMA- Δ OLIGO-GoLoco to perform (uncharacterized) functions other than its impact on the in vitro oligomerization assemblies. Therefore, I would also analyze the impact of NuMA- Δ OLIGO-GoLoco expression on the astral microtubules and dynein localization in such experimental settings.

We understand the Reviewer's point. Following his suggestions, we have imaged astral microtubules and cortical dynein in HeLa cells expressing NuMA-GoLoco and NuMA- Δ OLIGO-GoLoco upon treatment with MLN8237. α -tubulin staining of HeLa cells expressing NuMA-GFP constructs and treated with 50 mM MLN8237 showed that the partial inhibition of Aurora-A attained under these conditions does not perturb astral microtubules (see enclosed Figure 5). To assess the cortical levels of dynein in this experimental setting, we co-transfected HeLa cells with GFP-NuMA constructs and a FLAG-tagged version of dynein light intermediate chain 1 (LIC1), and imaged cortical LIC1 by FLAG immuno-staining (see enclosed Figure 6). The experiment shows that mild Aurora-A inhibition does not alter cortical recruitment of dynein in metaphase cells. Collectively, these additional analyses support the notion that only full-length NuMA but not NuMA- Δ OLIGO can rescue the misorientation phenotype observed upon Aurora-A inhibition, and that under this condition astral microtubules and cortical dynein recruitment are not affected.

Figure 5. Confocal x-y sections of mitotic HeLa cells expressing the indicated GFP-NuMA constructs, treated with DMSO or 50m mM MLN8237. Cells were stained with anti α -tubulin (white) and DAPI (blue). The distribution of the GFP signal in the same sections is shown on the right side. The reported sections have been chosen to visualize astral microtubules at best, and do not always include the spindle poles. Sections corresponding to cells treated with MLN and expressing GFP-NuMA or GFP-NuMA- Δ OLIGO-GoLoco show a single pole in focus because the spindle is misoriented.

Figure 6. Confocal x-y sections of mitotic HeLa cells co-transfected with FLAG-tagged dynein light intermediate chain 1 (FLAG-LIC1) and the indicated GFP-NuMA constructs, treated with DMSO or 50 mM MLN8237. Cells were stained with anti-FLAG (white) and DAPI (blue). The distribution of the GFP signal in the same sections is shown on the right

Minor points:

1. In general, the current draft does not provide the detailed information regarding the exact nature of the constructs in the Figure legends (whether GFP or mCherry-tagged). The readers need to move backward and forward from methods/main text to the Figure legends to check which plasmid/construct was utilized. For instance, in the Figure legend for Figure 5, the authors forgot to provide information related to the NuMA- Δ OLIGO-GoLoco that is the central construct for this piece of data.

We understand the point raised by the Reviewer. We had initially chosen to avoid repeating the kind of tag present on each construct to simplify the main text, and in the case of figure 5 we added a dedicated panel (panel 5a) to explain graphically the constructs, the tags and the boundaries of the NuMA and LGN domains used. To further smoothen the text, we have now modified the figure legends specifying always the nature of the LGN and NuMA tags, as suggested by the Reviewer.

2. The title of the authors is quite misleading since to my view it is not established whether in the cells under native conditions endogenous NuMA and LGN make Hexameric structures that are required for dynein-dependent spindle orientation. Therefore, as explained earlier, I would suggest the authors to evaluate this more critically by performing experiments (biochemically or cell biology-based) to test this using cellular assays.

We understand the comment. We think the additional experiments that we performed to revise the manuscript support the notion that NuMA and LGN form hexamers in cells that are essential for spindle orientation. Specifically, co-IP experiments with double-tagged LGN constructs presented in the new figure 4a demonstrate that LGN and NuMA oligomerize in mitotic cells and that the interfaces of this oligomers are the same of the hetero-hexamers that we characterize in vitro. We also show that the oligomerization deficient LGN- Δ OLIGO mutant induces spindle misorientation in Caco-2 cysts (new figure 3d and supplementary figure 3c). Therefore we think we provided evidence for the key role of NuMA:LGN hexamers in planar cell division. However, we do not have formal prove of how these hexamers spatially organize dynein to position the mitotic spindle. Therefore, we have modified the title to account for the Reviewer's observation, as suggested.

3. In experiments performed with Caco-2 cells, I would suggest using an alternate apical marker (e.g. ZO-1).

Following the Reviewer suggestion, we have stained with ZO-1 a set of Caco-2 cysts grown with wild-type cells or cells depleted of endogenous LGN and expressing the mCherry-tagged LGN rescue constructs. Representative images are shown in the enclosed figure 7. The localization of ZO-1 is consistent with the phalloidin staining presented in figure 3b and Supplementary figure 3c, and show that the single apical lumen is lost upon LGN ablation, and restored only the cysts expressing the wild-type LGN but not in the ones expressing LGN- Δ OLIGO.

Figure 7. Localization of the apical marker ZO-1 in Caco-2 cysts. Confocal sections of equatorial region of Caco-2 cysts grown from wild-type cells (left) or cells lacking LGN and stably expressing the rescue constructs LGN-WT-mCherry and LGN-ΔOLIGO-mCherry. Cysts were grown with the protocol used for figure 3, and stained with ZO-1 antibodies (Thermo Fisher Scientific, cat. 40-2200). DNA is visualized with DAPI (blue).

4. In the Western blot on Supplementary Figure 3a and 3, it is not clear which band represent LGN since other bands are also affected by RNAi-mediated depletion.

We thank the Reviewer for the observation. To get a cleaner LGN signal in the immuno-blots, we have purified a new batch of the monoclonal antibody against LGN, and repeated the blots presented in Supplementary figure 3a and 3b. As shown in the new panels, the specificity of the LGN signal improved significantly compared to the first submission, and only minor degradations of the LGN-mCherry constructs are detected beside the bands of LGN.

5. In S3g the cell expressing NuMAΔLGNBD has tilted spindle, and that is why the spindle-poles are not visible that could affect spindle pole quantification. I would suggest quantifying spindle-poles in cells where the spindle-poles are clearly visible.

The Reviewer is correct in saying that cells expressing the NuMA-ΔLGNBD constructs have tilted spindles and therefore only one pole at the time can be visualized properly in x-y confocal sections as the ones presented in Supplementary figure 4e (corresponding to Supplementary figure 3g in the first submission). For this reason, for the histogram of Supplementary figure 4f we have quantified only the mCherry signal of the spindle pole that was in focus in each cell image considered, and also for NuMA-WT we quantify one pole per image. We have modified the corresponding Method section and legend to specify this information that was not mentioned before.

Reviewer #3 (Remarks to the Author):

This paper addresses how the NuMA regulator LGN assembles complexes on the cortex that are structurally functional for spindle orientation. The foundation of the paper is the finding that fragments of NuMA larger than those previously studied interact with LGN form a 3:3 complex

between the proteins, as opposed to the 1:1 complex of the smaller fragment. A low resolution structure of the larger complex, determined by x-ray crystallography, is included in the paper. The structure, along with deletion analysis, allowed the authors to create mutations that selective disrupt the oligomerization function and they find that LGN proteins harboring these mutations are unable to rescue the spindle orientation defect found in cells that lack LGN. Overall this is an important contribution because it identifies a new LGN-NuMA contact point and furthers our understanding of the stoichiometry of the interaction and its functional importance. While it does leave the question of why oligomerization is required unanswered, hopefully this will be a topic for further investigation.

Minor comments:

Please label the dot plots and bar graphs through the paper (e.g. Figure 3c, e, ...) so that the relevant condition is underneath each plot.

We thank the Reviewer for the observation. We have better labeled all the plots specifying the condition under each bar and dot-plot, as suggested.

The following statement is incorrect unless the attached reference demonstrated that targeting full length NuMA to the cortex is sufficient for spindle orientation: "light-induced ectopic delivery of this NuMA fragment to the cortex results in dynein/dynactin recruitment but cannot support spindle pulling⁹, implying that during spindle orientation additional functions encoded by the C-terminal cargo binding portion of NuMA are essential for the spindle orientation process."

We thank the Reviewer for the comment. Indeed, Okumura and colleagues have shown that optogenetic targeting of full-length NuMA to the mitotic cortex of HeLa cells is sufficient for dynein-dynactin recruitment and spindle displacement (Okumura, eLife 2018, figure 1). Conversely, targeting with the same system of the NuMA fragment spanning residues 1-705 results in cortical dynein/dynactin recruitment in 9/10 of the cases, but cannot promote spindle displacement toward the cortex (Okamura, figure 4c). Our interpretation of this evidence is that the NuMA region comprising residues 706-2115 plays a role in spindle displacement. The Reviewer is correct in saying that the region of NuMA encompassing residues 706-2115 contains both a large portion of coiled-coil (up to about residue 1700) and the cargo binding portion, and that our results imply that both the dimerizing coiled-coil of NuMA and the cargo-binding portion containing the LGN-binding domain are important for MT-pulling forces. Therefore, we have rewritten the sentence to better clarify this point.

Reviewers' Comments:

Reviewer #1:

Remarks to the Author:

The authors have addressed all my concerns (and those of the other reviewers). I believe the manuscript is now ready for publication.

Reviewer #2:

Remarks to the Author:

The authors have substantially improved their manuscript, and I am mostly satisfied with their effort. However, I like them to clarify two points that I have with the revised manuscript. I am quite intrigued by the fact that overexpression of LGN Δ Oligo can also trigger spindle rocking to the same extent as upon overexpression of LGN wild-type construct (Figure 4; response to the reviewers). This data indicates that oligomerisation may not be crucial if excess LGN is present at the cell cortex for sufficient pulling forces. This is an essential piece of data and, therefore, it must be added to the manuscript in my view. As the authors know that in many cases, for instance, inactivation PIK1 or Cdk1 activity can enrich LGN/NuMA at the cell cortex, and in those cases, it may well be that non-oligomeric form of NuMA/LGN is capable of generation of pulling forces, and I think this should be discussed.

Additionally, in the Supplementary Figure 4, the authors have shown that NuMA Δ Oligo accumulates less at the poles than the full-length NuMA and have mentioned that 'implying that the inability of these constructs to rescue misorientation might be due to impaired oligomerization with LGN but also to spindle assembly defects (page 10)'. Authors have used NuMA Δ Oligo in the context of main Figure 5b and c, and it appears to me that expression of GFP-NuMA Δ Oligo-GoLoco shows weak NuMA localisation at the poles (even in cells treated with Aurora A inhibitor MLN8237) as well as chromosome instability defect. Therefore, one can argue that the spindle orientation defect seen upon expression of GFP-NuMA Δ Oligo-GoLoco in the presence of MLN8237 could be related to its impact on poles. Thus, it would be critical to check the effect of GFP-NuMA-GoLoco and GFP-NuMA Δ Oligo-GoLoco on spindle rocking as in Figure 4 (response to the reviewers) to know if NuMA Δ Oligo impacts spindle orientation when tethered to the cell cortex simply by not generating enough pulling forces, rather than its indirect impact on the spindle poles.

Reviewers' comments:

Rev#1

The authors have addressed all my concerns (and those of the other reviewers). I believe the manuscript is now ready for publication.

Rev#2

The authors have substantially improved their manuscript, and I am mostly satisfied with their effort. However, I like them to clarify two points that I have with the revised manuscript. I am quite intrigued by the fact that overexpression of LGN Δ Oligo can also trigger spindle rocking to the same extent as upon overexpression of LGN wild-type construct (Figure 4; response to the reviewers). This data indicates that oligomerisation may not be crucial if excess LGN is present at the cell cortex for sufficient pulling forces. This is an essential piece of data and, therefore, it must be added to the manuscript in my view. As the authors know that in many cases, for instance, inactivation Plk1 or Cdk1 activity can enrich LGN/NuMA at the cell cortex, and in those cases, it

may well be that non-oligomeric form of NuMA/LGN is capable of generation of pulling forces, and I think this should be discussed.

We thank the Reviewer for the comments. The rocking experiment suggests that great over-expression of LGN can induce aberrant pulling forces and defects in stabilization of the spindle position, regardless of the LGN/NuMA oligomerization. We initially included this evidence only in the rebuttal letter as we think that these conditions are not physiological because the LGN cortical levels are much higher than in wild-type cells. As suggested by the Reviewer, we have moved the rocking experiment in Supplementary Figure 3f-g, and discussed in the text these results in light of other conditions under which cortical levels of LGN and NuMA are abnormally increased.

Additionally, in the Supplementary Figure 4, the authors have shown that NuMA Δ Oligo accumulates less at the poles than the full-length NuMA and have mentioned that ‘implying that the inability of these constructs to rescue misorientation might be due to impaired oligomerization with LGN but also to spindle assembly defects (page 10)’. Authors have used NuMA Δ Oligo in the context of main Figure 5b and c, and it appears to me that expression of GFP-NuMA Δ Oligo-GoLoco shows weak NuMA localisation at the poles (even in cells treated with Aurora A inhibitor MLN8237) as well as chromosome instability defect. Therefore, one can argue that the spindle orientation defect seen upon expression of GFP-NuMA Δ Oligo-GoLoco in the presence of MLN8237 could be related to its impact on poles. Thus, it would be critical to check the effect of GFP-NuMA-GoLoco and GFP-NuMA Δ Oligo-GoLoco on spindle rocking as in Figure 4 (response to the reviewers) to know if NuMA Δ Oligo impacts spindle orientation when tethered to the cell cortex simply by not generating enough pulling forces, rather than its indirect impact on the spindle poles.

We thank the Reviewer for this observation. The experiments presented in Supplementary Figure 4 are conducted in HeLa cells depleted of endogenous NuMA and expressing the NuMA Δ Oligo construct. Quantification of the levels of the NuMA Δ Oligo at the spindle poles revealed that it accumulates less at the poles than NuMA-WT, and therefore it is not possible to ascribe the misorientation phenotype observed in sh-NuMA cells expressing NuMA Δ Oligo to its impaired cortical functions rather than to its spindle pole defects.

On the other hands, experiments of Figure 5b-c are carried out in HeLa cells retaining endogenous NuMA and expressing the NuMA-GoLoco constructs (either WT or Δ Oligo). Upon treatment of these cells with the Aurora A inhibitor MLN8237, endogenous NuMA accumulates at the spindle poles and assembles a correct spindle, as documented in Gallini et al. *Curr. Biol.* 2016, and shown in Figure 5 previous-response to the Reviewers, where we stained cells with α -tubulin and visualized the existence of proper astral microtubules. In these MLN8237-treated cells, NuMA-GoLoco proteins go to the cortex bypassing the MLN8237 treatment. Therefore we can say that the results of these spindle orientation experiments reflect genuinely the functions of NuMA-GoLoco:LGN complexes at the cortex, uncoupled from the activities of NuMA at the poles and from endogenous cortical NuMA.

We cannot completely rule out that the construct NuMA- Δ Oligo-GoLoco has a dominant-negative function at the poles, although the evidence that it accumulates less than NuMA-WT-GoLoco and that no gain-of-function has been described for this mutant so far, makes this possibility unlikely.

The fact that in some panels of the Figure 5 previous-response to the Reviewers the levels of GFP are low at the poles is because the confocal planes displayed were selected to show astral microtubules at best, and do not always include the spindle poles.

Based on all these considerations, we do not think that performing the rocking experiments with NuMA-GoLoco constructs in MLN8237-treated cells would provide novel information on the properties of the NuMA-GoLoco constructs at the cortex, or on the role of NuMA-LGN oligomerization in spindle placement. Therefore, we would prefer not to include these experiments in the manuscript.